# Low power flexible monolayer MoS$_2$ integrated circuits

Jian Tang [1,2,7], Qinqin Wang[1,2,7], Jinpeng Tian[1,2,7], Xiaomei Li[1,2,3], Na Li[1,4], Yalin Peng[1,2], Xiuzhen Li[1,2], Yanchong Zhao [1,2], Congli He[5], Shuyu Wu[6], Jiawei Li[1,2], Yutuo Guo[1,2], Biying Huang[1,2], Yanbang Chu[1,2], Yiru Ji[1,2], Dashan Shang [6], Luojun Du[1,2], Rong Yang [1,4], Wei Yang [1,2,4], Xuedong Bai [1,2], Dongxia Shi[1,2] & Guangyu Zhang [1,2,4] ✉

Monolayer molybdenum disulfide (ML-MoS$_2$) is an emergent two-dimensional (2D) semiconductor holding potential for flexible integrated circuits (ICs). The most important demands for the application of such ML-MoS$_2$ ICs are low power consumption and high performance. However, these are currently challenging to satisfy due to limitations in the material quality and device fabrication technology. In this work, we develop an ultra-thin high-κ dielectric/metal gate fabrication technique for the realization of thin film transistors based on high-quality wafer scale ML-MoS$_2$ on both rigid and flexible substrates. The rigid devices can be operated in the deep-subthreshold regime with low power consumption and show negligible hysteresis, sharp subthreshold slope, high current density, and ultra-low leakage currents. Moreover, we realize fully functional large-scale flexible ICs operating at voltages below 1 V. Our process could represent a key step towards using energy-efficient flexible ML-MoS$_2$ ICs in portable, wearable, and implantable electronics.

Flexible electronics plays an integral role in a large spectrum of fields including information technology, energy generation and storage, biosensing, and diagnosis[1–5]. Among them, flexible integrated circuits (ICs) dealing with information processing are favorable in portable, wearable, and implantable electronics with technological demands towards flexibility and robustness of large-area devices[6–8]. Conventional flexible ICs are usually fabricated from organic semiconductors[6,8], silicon of either amorphous or polycrystalline forms[9,10], oxide semiconductors[11,12], and carbon nanotubes (CNTs)[13–17] via the thin-film-transistor (TFT) technology. Recently, the 2D semiconductor of monolayer MoS$_2$ (ML-MoS$_2$) emerged as an advanced channel material in large-area flexible TFTs[18–26]. In principle, such TFTs have great potential in both high performance and low-power applications, if considering the following merits. First, ML-MoS$_2$ is atomically thin

(only ~0.7 nm) and smooth yet mechanically strong (in-plane) and flexible (out-of-plane)[1,27]. Such a thin channel also offers benefits in ultra-scale devices where short channel effects would be the main concern[28–30]. Second, its 2H phase has a moderate band gap of ~2 eV, between that of silicon (~1.1 eV) and indium gallium zinc oxide (~3.5 eV), hence can work with both low off-state and high on-state currents[31,32]. Third, it has high electrical quality and is available at wafer scale[22,33–36].

It has been shown that, at room temperature, rigid ML-MoS$_2$ TFTs can feature high electron mobility ($\mu$) of >100 cm$^2$·V$^{-1}$·s$^{-1}$, high on/off ratio of >10$^8$, low subthreshold swing (SS) approaching the thermionic limit of 60 mV·dec$^{-1}$, and high on-current of ~1.1 mA·μm$^{-1}$ at a supply voltage of 1.5 V[37–39]. Hence, one might ask if such TFT technologies developed on rigid substrates could be transferred onto flexible substrates and be superior to existing flexible TFT technologies. Up to

[1]Beijing National Laboratory for Condensed Matter Physics and Institute of Physics, Chinese Academy of Sciences, Beijing 100190, China. [2]School of Physical Sciences, University of Chinese Academy of Sciences, Beijing 100190, China. [3]Shanghai Key Laboratory of Multidimensional Information Processing, East China Normal University, Shanghai, China. [4]Songshan Lake Materials Laboratory, Dongguan 523808, China. [5]Institute of Advanced Materials, Beijing Normal University, Beijing 100875, China. [6]Institute of Microelectronics, Chinese Academy of Sciences, Beijing 100029, China. [7]These authors contributed equally: Jian Tang, Qinqin Wang, Jinpeng Tian. ✉e-mail: gyzhang@iphy.ac.cn

now, large-scale flexible ML-MoS$_2$ ICs with various fully functional units such as logic gates, inverters, and ring oscillators (ROs) have been demonstrated vividly[23–25]. However, limited by materials and fabrications, such large-area flexible ICs suffer from either low performances or high-power consumptions, and it is challenging to realize both in a synergic manner.

In this work, we demonstrate the realization of low-power flexible ICs based on ML-MoS$_2$. The technological advances of this work lay on both the newly developed high-quality ML-MoS$_2$ wafers and the ultra-thin high-κ dielectric/metal-gate technology. High-quality 4-inch wafer-scale ML-MoS$_2$ films are epitaxially grown on sapphire via an oxygen-assisted chemical vapor deposition (CVD) approach we developed recently[34]. As-grown films are strictly monolayer, uniform across entire substrates, and have large domain sizes over 200 μm. Such high-quality materials offer a desirable choice for realizing large-scale flexible ICs. Here, the flexible ICs are fabricated via a gate-first technology in which ultra-thin high-κ dielectrics are deposited on gate electrodes to achieve a record-low equivalent oxide thickness (EOT). Due to the enhanced gating efficiency, our ML-MoS$_2$ flexible TFTs can work under operation voltages below 1 V.

## Results

### Gate-first technology for ultra-thin high-κ dielectric deposition

In a power-efficient system, transistors and circuits should ideally have a low-driving voltage which can be created by using high-capacitance dielectric layers for efficient electrostatic gating. We thus employed ultra-thin high-κ dielectrics of HfO$_2$, a technique widely used in the present silicon-based high-performance and low-power electronics. Since ultra-thin high-κ dielectrics are very difficult to deposit on MoS$_2$ due to the lack of surface dangling bonds[40,41], we hence developed the high-κ dielectric/metal-gate technology, i.e. a gate-first technology, for ML-MoS$_2$ TFTs. Figure 1a illustrates the device geometry with buried Ti-Au-Ti as the local back-gate electrode, ultra-thin HfO$_2$ as the dielectric layer, ML-MoS$_2$ as the channel, and Au as the source/drain electrodes.

During the device gate-first fabrication process, buried-gate electrodes of Ti-Au-Ti (1-5-1 nm) were first deposited on a substrate (either rigid or flexible) by standard lithography and electron beam evaporation processes. Note that the top 1 nm-Ti layer after oxidation in O$_2$ plasma serves as the seeding layer for following atomic layer deposition (ALD) of the high-κ dielectrics; while the bottom 1 nm-Ti layer acts as the adhesive layer between Au and substrate. Other than the deposition of the seeding layer for the subsequent HfO$_2$ deposition on 2D materials like graphene[42], our approach employs the seeding layer on metal gates which are more general and compatible with the semiconductor fabrication process. Before and after metal gate depositions, we performed oxygen plasma cleaning to remove the photoresist residues on side walls and top surface of patterns/metal-gates introduced from the lithography process, as illustrated in Supplementary Fig. 1. With the help of this cleaning process, we can produce flat metal gates with clean surfaces and sharp boundaries. After ALD, the deposited HfO$_2$ layers on metal gates are very uniform, as

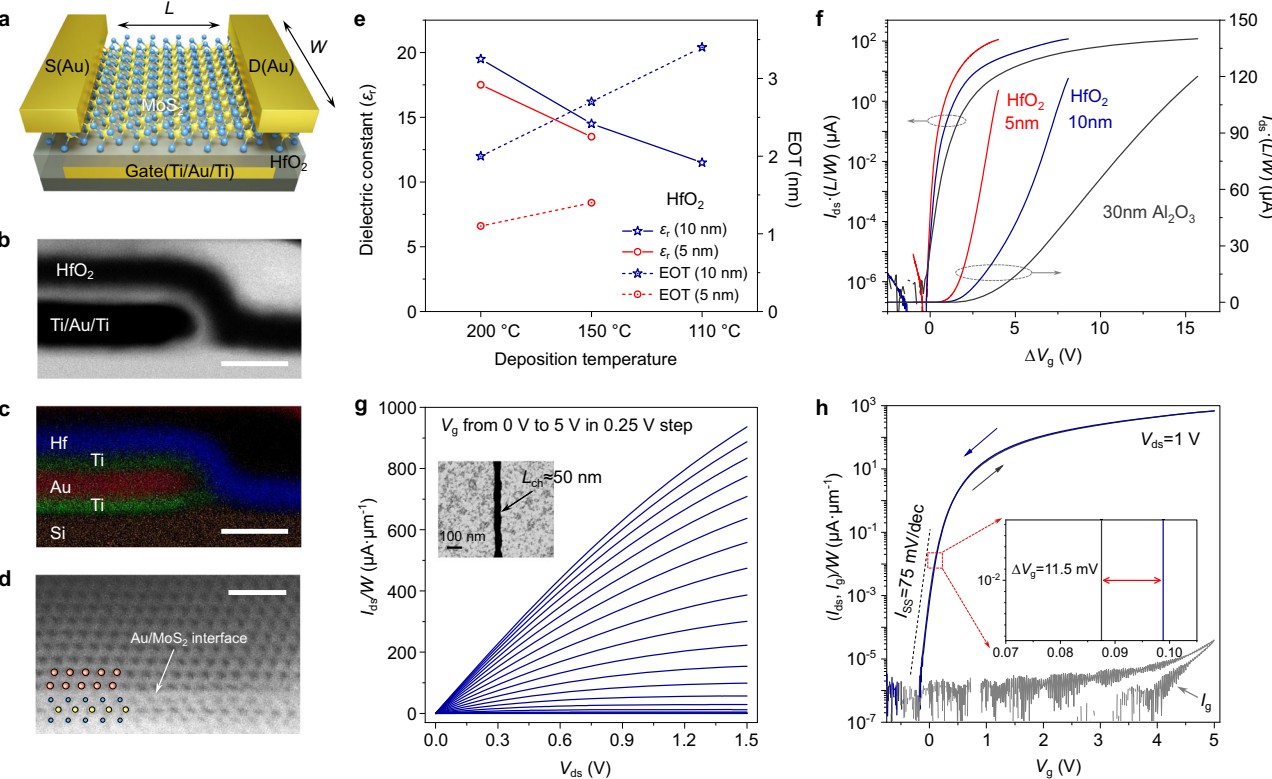

**Fig. 1 | Characterizations of monolayer MoS$_2$ thin-film-transistor (TFTs) on SiO$_2$ substrates. a** Schematic illustration of a buried-gate MoS$_2$ TFT. **b, c** Cross-sectional scanning transmission electron microscope (STEM) image (**b**) of 5-nm HfO$_2$ on the buried Ti-Au-Ti (1-5-1 nm) gate electrode, and corresponding energy disperse spectroscopy (EDS) elemental mapping image (**c**) clearly shows the distribution of Au, Ti, Hf and Si elements. Scale bars, 10 nm. **d** Atomic-resolution cross-sectional STEM image of Au-MoS$_2$ contact interface. Red, yellow, and blue spheres are Au, Mo, and S atoms, respectively. Scale bar, 1 nm. **e** The dielectric constant and equivalent oxide thickness (EOT) of HfO$_2$ layers deposited at different temperatures. **f** Normalized transfer curves of MoS$_2$ TFTs ($L$ = 5 μm, $W$ = 40 μm) at $V_{ds}$ = 1 V with 5 nm, 10 nm HfO$_2$ and 30 nm Al$_2$O$_3$ as the dielectric layer, individually. **g, h** Typical electrical output and transfer curves of MoS$_2$ TFTs with $L_{ch}$ ≈ 50 nm and 5-nm HfO$_2$ as the dielectric layer. Inset to (**g**) is the scanning electron microscope (SEM) image of the $L_{ch}$ ≈ 50 nm device. The $I_{ds}/W$-$V_g$ curve features with the sharp subthreshold swing ($I_{SS}$ = 75 mV·dec$^{-1}$, refer to the black dash line) and small hysteresis ($\Delta V_g$ = 11.5 mV, inset to **h**). The leakage current ($I_g$) is represented in gray color in (**h**).

characterized by the atomic force microscope (AFM). The measured surface roughness of 5-nm $HfO_2$ deposited on Ti/Au/Ti metal gates is typically <0.5 nm. After high-κ dielectric deposition, a 4''-wafer-sized ML-MoS$_2$ film (See Supplementary Fig. 2 and Table 1 for film quality characterizations and comparisons) was transferred onto the overall structure by a wet chemical etching, and the transferred ML-MoS$_2$ films are almost intact and flat on target substrates process (see Methods and Supplementary Fig. 3). Finally, multistep lithography, reactive ion etching (RIE), and electron beam evaporation were performed to define the channel and source/drain (S/D) contact regions.

For a typical TFT on the $SiO_2$ substrate, we characterized its interfaces between different layers by cross-sectional scanning transmission electron microscope (STEM). Figure 1b, c shows a typical STEM image and corresponding Au, Ti, Hf, Si elemental mapping from energy disperse spectroscopy (EDS) at the $HfO_2$/metal-gate interface. We can see a uniform and conformal coating of $HfO_2$ with a thickness of 5 nm on the metal gate. Figure 1d shows an atomic-resolution STEM image of the Au-MoS$_2$ interface. The sharp interface between periodically arranged Au atoms and three-atom-thick MoS$_2$ surface without any defects or cracks suggests the high quality of contacts.

To evaluate the dielectric properties of $HfO_2$ layers deposited on metal gates by ALD, we performed capacitance measurements (Supplementary Fig. 4). In Fig. 1e, we plot the dielectric constants ($\varepsilon_r$) of 5-nm and 10-nm thick $HfO_2$ layers deposited at 110, 150, and 200 °C. Obviously, higher deposition temperature and thicker thickness are beneficial to achieving a better dielectric property. Hence, the ALD temperature of 200 °C was applied for rigid devices; whereas it is 150 or 110 °C for flexible devices to reduce the temperature-induced substrate deformation. In our batch fabrication of devices, the minimum $HfO_2$ thickness ($t_{HfO2}$) is 5 nm on rigid substrates, e.g., $SiO_2$, and 10 nm on flexible substrates, e.g., polyethylene terephthalate (PET, see Supplementary Fig. 5 for roughness characterizations), for a reliable device yield. Note that the 5-nm $HfO_2$ layer has an effective oxide thickness (EOT) of only 1.1 nm. Such low EOT is beneficial to high device transconductance at low supply voltages and efficient gating of MoS$_2$ channels.

## ML-MoS$_2$ TFTs on rigid substrates

Firstly, let's evaluate ML-MoS$_2$ TFTs on $SiO_2$ substrates with normalized channel lengths ($L_{ch}$) and widths ($W$). Figure 1f shows the normalized transfer curves ($I_{ds}\cdot L/W \cdot \Delta V_g$) of devices with $HfO_2$ layer thickness of 5 and 10 nm. As a comparison, 30 nm thick $Al_2O_3$ devices are also included. It can be clearly seen that with decreased $t_{HfO2}$ and increased $\varepsilon_r$, the gate voltages ($V_g$) can be reduced from 15 V to 3 V and the SS can be reduced from 250 mV·dec$^{-1}$ to 75 mV·dec$^{-1}$, while preserving the on-state current ($I_{on}$) densities and on/off ratios.

Figure 1g, h demonstrate typical output ($I_{ds}/W \cdot V_{ds}$) and transfer curves ($I_{ds}/W \cdot V_g$) of a short channel device with $t_{HfO2} = 5$ nm and $L_{ch} \approx 50$ nm. This device features ultra-high on/off ratio of ~$10^9$, sharp subthreshold slope (SS) of $I_{SS} = 75$ mV·dec$^{-1}$ over 5 orders of magnitude, and negligible hysteresis ($\Delta V_g$ ~ 11.5 mV, inset to Fig. 1h) subjected to the high quality of MoS$_2$/HfO$_2$ interface. The ultra-high on/off ratio could guarantee an ultra-low static power dissipation. When devices work at a stand-by mode (off state), the leakage current ($I_g/W$) can reach below 1 pA·μm$^{-1}$ (Fig. 1h) for a general $W/L_{ch} = 5$ μm/50 nm device. Due to the high gating efficiency, the maximum field could approach ~1 V·nm$^{-1}$ and the carrier density can approach $n_i = 5.15 \times 10^{13}$ cm$^{-2}$ at $V_g = 5$ V. Note that the effective carrier densities in MoS$_2$ channel would be overestimated based on the metal-insulator-metal capacitance (MIMCAP) results, thus we carried out metal-oxide-semiconductor capacitance (MOSCAP) measurements based on the MoS$_2$ FET (Supplementary Fig. 6). Such high carrier densities in ML-MoS$_2$ channels are among the highest in previous reports. As a result, the maximum on-state current densities ($I_{on}/W$) could reach 936 μA·μm$^{-1}$ at $V_{ds} = 1.5$ V with $L_{ch} = 50$ nm in Fig. 1g, which is

comparable to the highest value (1135 μA·μm$^{-1}$ at $V_{ds} = 1.5$ V) previously achieved in Bi-contacted ML-MoS$_2$ TFTs with $L_{ch} = 35$ nm[39]. More data on MoS$_2$ TFTs with $L_{ch}$ of 150 nm and 300 nm appears in Supplementary Fig. 7. Note that even Au-contacts show slightly higher contact resistances than Bi-contacts, our devices still show linear output characteristics at small bias and saturated output behaviors at high bias voltages, which is important for the output current capacity of short channel devices. Besides, $I_{on}/W$ is ~720 μA·μm$^{-1}$ at $V_{ds} = 1$ V, meeting the low-power application metrics of the International Roadmap for Devices and Systems (IRDS, 2024)[43].

## ML-MoS$_2$ TFTs on flexible substrates

Next, we implement the technique of ultra-thin high-κ dielectric/ metal gate deposition on flexible PET substrate. Figure 2a shows 4'' wafer-scale ML-MoS$_2$ TFTs on PET substrate. Now let's evaluate devices on flexible substrates based on 10 nm $HfO_2$. These long-channel TFTs have a high device yield (>96%) and nice spatial uniformity. In Fig. 2b we show transfer curves of 500 randomly picked TFTs with $L_{ch}$ varying from 5 to 75 μm. Statistics on the device's $\mu_{FE}$, on/off ratio, threshold voltage ($V_{th}$), and SS are shown in Fig. 2c. According to Lorentz distribution fittings, $\mu_{FE}$ averages at ~70 cm$^2$·V$^{-1}$·s$^{-1}$ (maximizes at >110 cm$^2$·V$^{-1}$·s$^{-1}$); on/off ratio averages at $5 \times 10^7$ (maximizes at ~$1 \times 10^9$); $V_{th}$ is centered at 0.96 ± 0.4 V; and SS averages at 83 mV·dec$^{-1}$. All these performances are comparable to those previously achieved in rigid ML-MoS$_2$ TFTs (Supplementary Table 2) and greatly improved over the previous state-of-the-art flexible TFTs (Supplementary Table 3). It is worth noting here that both the positive $V_{th}$ and small SS are critical for the low power consumption in our devices, as will be shown later. Note that we also realize the fabrications of flexible ML-MoS$_2$ FETs based on 5 nm $HfO_2$ as measured in Supplementary Figs. 8, 9. Considering the low device yield (~60%) and higher leakage current level of the 5 nm $HfO_2$, we mainly adopt 10 nm $HfO_2$ as dielectric layer for integrated logic gates in following experiments.

According to previous studies[44], the $V_{th}$ variability comes mainly from sources including material uniformity (defects, grain boundaries, and layer thicknesses), oxide roughness, surface (MoS$_2$ surface), and interface (contact interface, oxide-MoS$_2$ interface) cleanness. In this study, MoS$_2$ is strictly monolayer without additional layers and of high quality. High defects (mainly S-vacancies) density could cause severe n-doping in the devices, shifting $V_{th}$ to very negative values. In present devices, the positive $V_{th}$ reflects the low density of vacancy defects in the MoS$_2$ channel region. The domain sizes in our samples are over 200 μm, much larger than the channel sizes, suggesting that grain boundaries are also not likely the source of $V_{th}$ variations. The thin $HfO_2$ layers deposited on metal gates are smooth with a surface roughness of ~0.5 nm, measured by AFM. We thus conclude that the main source of $V_{th}$ variability, although it is small, comes mainly from the surface and interface cleanness.

The high device performance is attributed to several aspects. First, our ML-MoS$_2$ films are of high quality. The extracted sheet resistance $R_\square$ is ~5.2 kΩ at $n_i \approx 2.8 \times 10^{13}$ cm$^{-2}$ fitted by the transfer length method (TLM), as shown in Fig. 2d. Second, the employment of $HfO_2$ dielectrics allows us to tune carrier densities in MoS$_2$ over than $5 \times 10^{13}$ cm$^{-2}$. Third, the contact resistance $R_c$ between Au and MoS$_2$ is as low as ~0.59 kΩ·μm (see Fig. 2d and Supplementary Fig. 10 for the $R_c$ extraction based on long channel and short channel devices). Such low $R_c$ is facilitated by the ultra-slow Au-deposition rates (see Methods) and the bottom gate structure in which the contact region of ML-MoS$_2$ can be doped to be metallic properties. Based on our data from both rigid and flexible TFTs, here we highlight the on-state current densities ($I_{ds}/W$) as a function of $L_{ch}$ in Fig. 2e. Clearly, our data points lay at the upper envelope boundary if compared with literature works from monolayer flexible or rigid MoS$_2$ TFTs (all reference data points acquired from Supplementary Tables 2, 3).

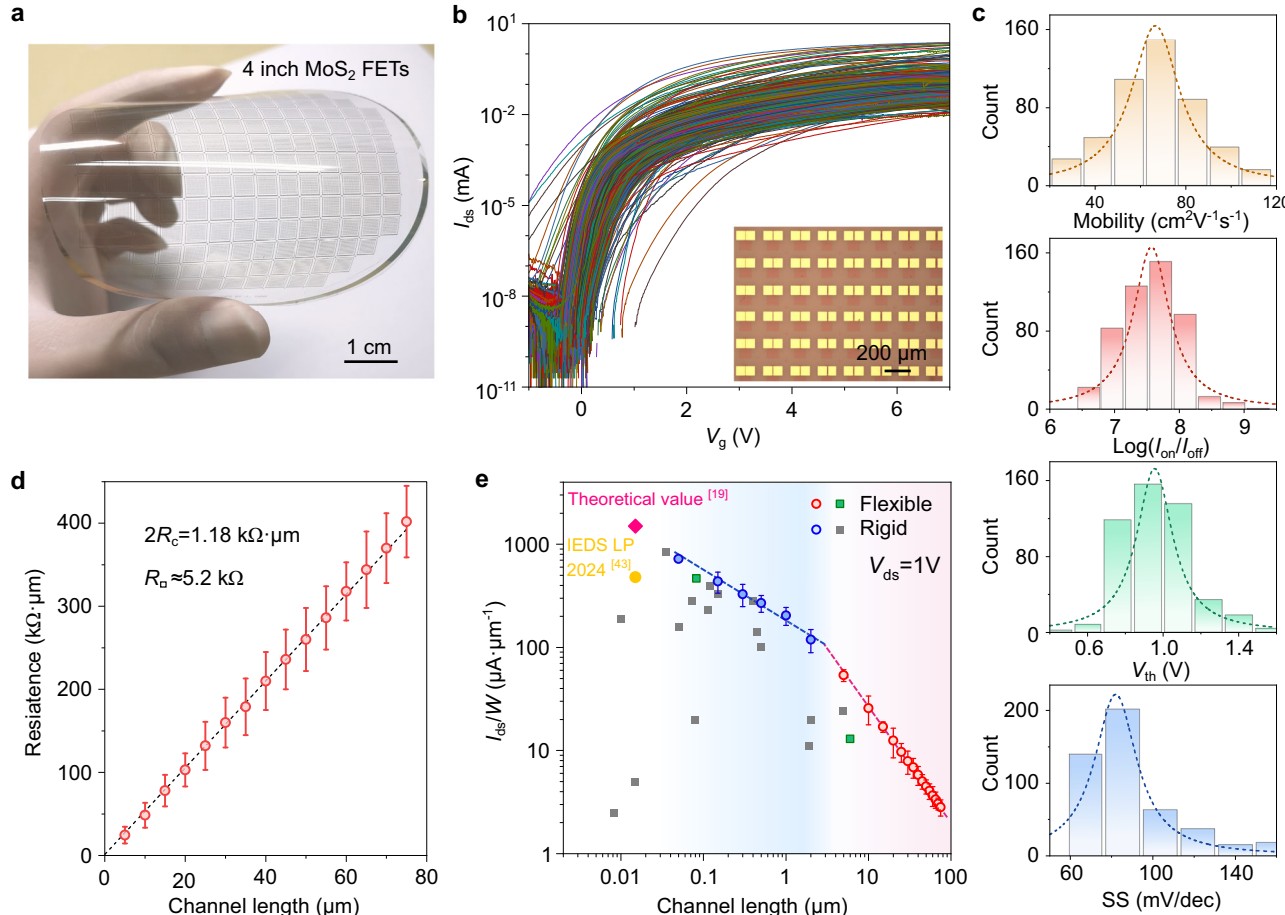

**Fig. 2 | Flexible ML-MoS$_2$ TFTs and electrical properties. a** Photograph of an as-fabricated 4-inch MoS$_2$ TFTs on the polyethylene terephthalate (PET) substrate. Scale bar, 1 cm. **b** Transfer curves of 500 randomly picked TFTs with the 10 nm HfO$_2$ dielectrics. Bias voltage ($V_{ds}$) is 1 V, and the device yield is 96.2%. Inset to (**b**) shows the enlarged image of the device array. Scale bar, 200 μm. **c** Histogram and Lorentz distribution fits of the device mobility, on/off ratio, threshold voltage and sub-threshold swing extracted from the transfer curves in (**b**). **d** Contact and sheet resistance measurements from transfer length method (TLM) at $n_i \approx 2.8 \times 10^{13}$ cm$^{-2}$. **e** Statistics and comparisons of the on-state current density of the flexible and rigid TFTs at $V_{ds} = 1$ V versus channel length. This work is highlighted in red and blue circles, while the reference data points are drawn in gray and green squares (please refer to Supplementary Tables 2, 3 for details). Error bars in (**d**) and (**e**) are taken from 10 transistors with the same channel length.

## Low-power and high-performance flexible ICs

We further fabricated large-area flexible ICs based on ML-MoS$_2$. Figure 3a shows a photograph of $4 \times 4$ cm$^2$ electronic circuits fabricated on the PET substrate. As logic gates and inverters are essential building blocks of ICs, we included logic inverters, NAND, NOR, and AND gates in this batch (Fig. 3b, c).

The output characteristic of ML-MoS$_2$ inverter shows abrupt switching behavior with power supply voltages $V_{dd} = 0.5$ V (Fig. 3d). The trip point of the inverter is when output voltage ($V_{out}$) equals input voltage ($V_{in}$) and of particular importance for low-power electronics as it determines the driving voltage of integrated logic circuits. The typical inverter also exhibits full swing output behavior with noise margin (NM) of $0.8 \times (V_{dd}/2)$ and voltage gain of 120. Note that the highest voltage gain is 192 at $V_{dd} = 0.5$ V and 397 at $V_{dd} = 1$ V, and it can approach 1000 (2670) at $V_{dd} = 2$ V (4 V), the highest ever achieved in MoS$_2$ inverters (see Supplementary Fig. 11). Note that the rail-to-rail operations could be realized by further optimizing the circuit design of the FET units[45], combing p-type FETs as building blocks[46] or introducing doping techniques[47]. As can be seen in Fig. 3e, our MoS$_2$ inverters have ultra-high gains and reliable outputs within the sub-1V supply voltage zone, as compared with other flexible inverters or rigid inverters based on 2D materials (the reference data points acquired from Supplementary Table 4). The ultra-high voltage gain is enabled by the strong gate controllability, while the trip point is mainly

determined by the $V_{th}$ distribution in our MoS$_2$ TFTs. We also analyzed the individual TFT unit within an inverter (Supplementary Fig. 12) with intrinsic gains[48] (defined as $A_i = g_m/g_d$) from $10^3$ to $10^4$, where $g_m$ and $g_d$ are the transconductance and output conductance of the TFT. Through monitoring the channel current of an inverter, the MoS$_2$ TFT unit works mainly at the deep-subthreshold regime with ultra-low current densities. The calculated maximum output power of an inverter is 10.3 pW·μm$^{-1}$, at the same level as an oxide semiconductor inverter[32]; and the quiescent power is below 0.1 pW·μm$^{-1}$ at $V_{dd} = 1$ V (see Supplementary Table 5 for power consumption comparisons).

Typical output characteristics of ML-MoS$_2$ logic NAND, NOR, AND gates are also shown in Fig. 3f. 0.5 V pulses of $V_A$ and $V_B$ with 1 s delay are used as input signals, where 0.5 V for the logic '1' and 0 V for the logic '0'. All logic gates have correct Boolean output functionalities with a low supply voltage $V_{dd} = 0.5$ V.

In addition to logic gates and inverters, flexible ML-MoS$_2$ ROs with various number of stages were also fabricated to demonstrate the reliability for low voltage operations. Figure 4a shows the optical microscope image of 11-stage ROs which integrate cascading eleven inverters and an additional inverter as the output buffer for measurements. The corresponding schematic circuit with three terminal electrodes of $V_{dd}$, $V_{ss}$, and $V_{out}$ is shown in Fig. 4b. Figure 4c shows the stable electrical output signals of an 11-stage RO at supply voltage $V_{dd}$ from 0.3 V to 1 V ($V_{ss}$ terminal is grounded). Such low-driving voltages

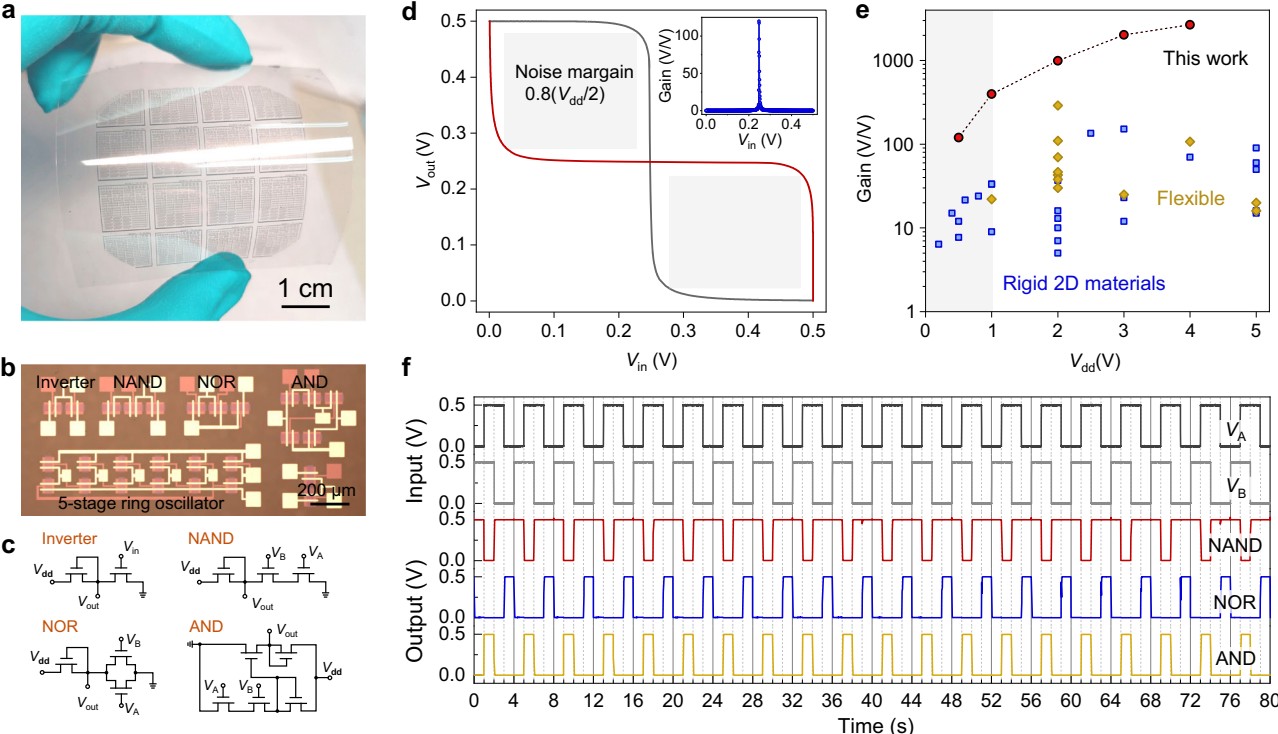

**Fig. 3 | Flexible integrated circuits at low driving voltages. a** Photograph of $4 \times 4\ cm^2$ integrated monolayer $MoS_2$ TFTs on the flexible PET substrate. Scale bar, 1 cm. **b**, **c** Optical image (**b**) and schematic circuits (**c**) of logic inverters, NAND, NOR, AND gates and 5-stage ring oscillators. Scale bar, 200 μm. $V_{dd}$, power supply voltage; $V_{in}$, $V_A$, or $V_B$, input voltages; $V_{out}$, output voltage. **d** Voltage transfer curves (VTCs) of a logic inverter with noise margin of $0.8(V_{dd}/2)$ at $V_{dd} = 0.5\ V$. Inset shows corresponding voltage gain of 120. **e** Comparison of the inverter voltage gain as a function of $V_{dd}$ among flexible inverters or rigid inverters based on 2D materials. The reference data points are listed in Supplementary Table 4. **f** Output characteristics of logic NAND, NOR, AND gates as a function of input voltage pulses with $V_{dd} = 0.5\ V$.

are facilitated from the small trip point of logic inverters. In contrast, if negative voltages are applied on the $V_{ss}$ terminal while the $V_{dd}$ terminal is grounded, the RO could output negative signals starting from $V_{ss} = -0.4\ V$, as the swing threshold voltages of logic inverters shift negatively with the decreasing the operation voltages of $V_{ss}$ or $V_{dd}$ terminal (Supplementary Fig. 11c, d).

Generally, the output amplitude increases with the supply voltages and the oscillation frequency (*f*) decreases with the number of stages. Figure 4d shows the maximum output oscillation signals of 3-, 5- and 11-stage ROs operated at $V_{dd} = 5\ V$. The ROs have reliable outputs with supply voltages varying from 0.3 V to 5 V (Fig. 4e). The maximum *f* is 24.8, 14.29, and 6.7 MHZ for 3-, 5- and 11-stage ROs, respectively. The corresponding propagation stage delay time ($\tau_{pd} = 1/2Nf$, where *N* is RO stage number) at $V_{dd} = 5\ V$ is 6.7, 7.0, and 6.8 ns for 3-, 5-, and 11-stage ROs, respectively. Compared with previous flexible ROs fabricated from other 2D materials, organic, oxides or CNTs, our ROs locate at the preferred conner with the low driving voltages (sub-1 V) and fast propagation stage delay time (Fig. 4f). Note that, in this work, the parasitic capacitance from the overlapping region between the contact electrodes/additional outer $MoS_2$ region with the gate is detrimental to high-frequency operation. The typical channel length/width of TFT unit is 1 μm/3 μm for ROs, and more device details and parasitic capacitance optimization is demonstrated in Supplementary Fig. 13. A typical real-time outputs of oscillating RO is included in the Supplementary Movie to show the stable operation.

Finally, we performed bending tests of the flexible ML-$MoS_2$ ICs. As an example, here we show results from flexible $MoS_2$ TFTs and 5-stage RO in Supplementary Figs. 14–17. On/off ratios as well as field-effect mobilities of individual TFTs could be well-preserved under a minimum bending radius (*R* = 2.4 mm). Under such bending, either along the X- or Y- direction, the 5-stage RO exhibits stable outputs

without obvious degradations. The excellent mechanical flexibility and endurance, reliable electrical outputs of ROs, and high device yield indicate the good device performances and uniformity of our devices fabricated on flexible substrates; and the lowest operation supply voltage indicates the potential for low-power electronic applications.

## Discussion

We developed an ultra-thin high-κ dielectric on metal gate technology for ML-$MoS_2$ TFTs. Benefiting from the high-quality ML-$MoS_2$ wafers as well as ohmic contacts, our ML-$MoS_2$ TFTs could be tuned to high carrier density of $5.15 \times 10^{13}\ cm^{-2}$ and allow a high current capacity of $936\ \mu A \cdot \mu m^{-1}$ at $V_{ds} = 1.5\ V$ with sharp SS of $75\ mV \cdot dec^{-1}$, positive threshold voltages, negligible hysteresis, and ultra-low leakage currents. Large-area flexible TFTs and ICs show excellent spatial uniformity and a high device yield of >96% with fully functional inverters, logic gates, and ring oscillators working reliably under supply voltages below 1 V. The power consumption of an inverter can be as low as $10.3\ pW \cdot \mu m^{-1}$ at $V_{dd} = 1\ V$. Besides, both the voltage gains of inverters and the propagation delay time of ROs we achieved are record-high values against previous flexible devices. Our results suggest that ML-$MoS_2$ is a very competitive channel material in flexible ICs for both high performance and low power applications.

## Methods

### Growth of high-quality ML-$MoS_2$ films

The growth was carried out in a CVD system using S (Alfa, 99.5%, 8 g) and $MoO_3$ (Alfa, 99.9995%, 30 mg) powder as reaction sources. 4-inch c-plane sapphire wafers were used as substrates. In order to achieve large domain sizes, we intended to reduce the nucleation density during the growth process and elongate the growth time accordingly. The nucleation density of $MoS_2$ was

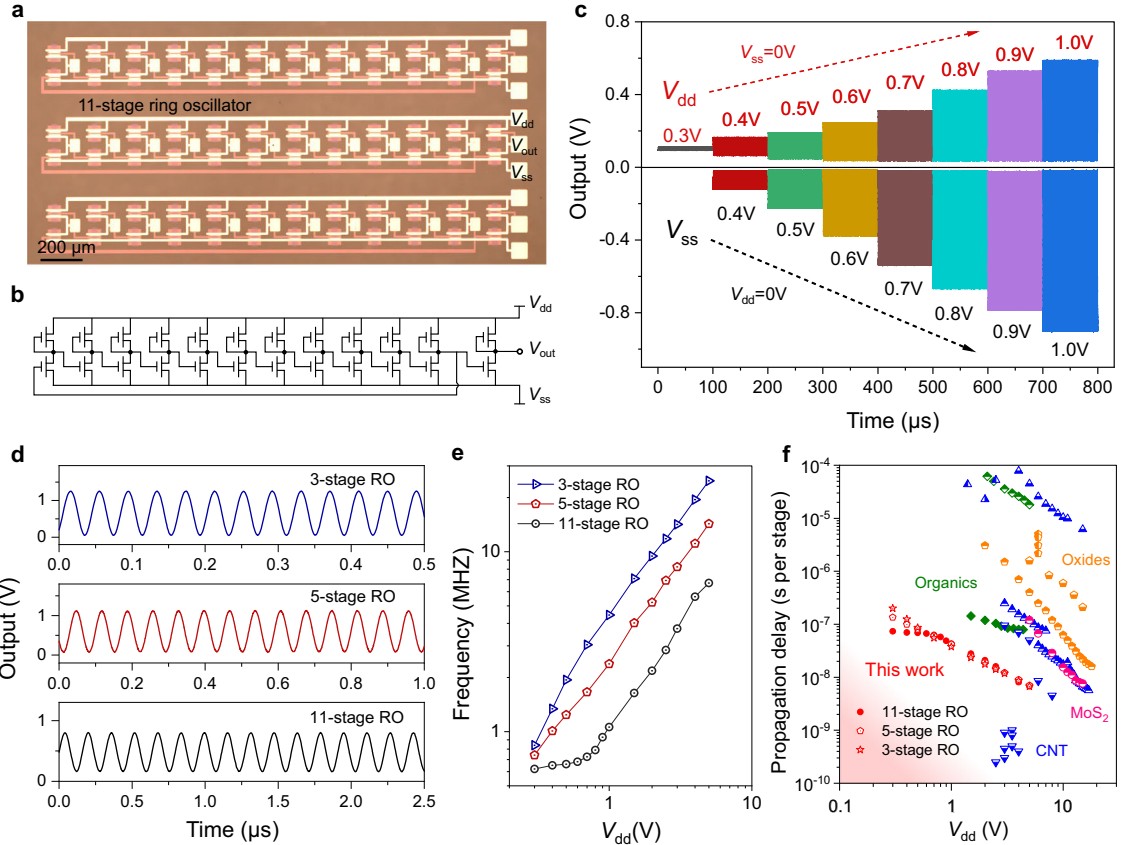

**Fig. 4 | Flexible ML-MoS$_2$ ring oscillators (ROs) operated at sub-1 V supply voltages. a**, **b** Optical image and schematic circuit diagram of 11-stage ROs. Scale bar, 200 μm. $V_{dd}$, or $V_{ss}$, power supply voltages applied on drain or source electrodes. **c** Output signals of an 11-stage RO operated with sub-1 V supply voltages from 0.3 V to 1 V. **d** Output signals of 3-stage, 5-stage and 11-stage ROs operated at $V_{dd}$ = 5 V. The output oscillation frequency increases by decreasing the stage number. **e** Summarized output frequencies of ROs as a function of $V_{dd}$ from 0.3 V to 5 V. The output oscillation frequency increases with supply voltages. **f** Comparisons of propagation stage delay time as well as supply voltages with literature works. Detailed parameters are listed in Supplementary Table 6.

reduced by using higher oxygen flow rate for MoO$_3$ source and reducing the Mo-source flux by lowering its evaporation temperature, increasing the distance between the Mo-source and sapphire substrate. During the growth, carrier gases of Ar (40 sccm) and Ar (240 sccm)/O$_2$ (10 sccm) were fluxed for S power and MoO$_3$ individually and the pressure in the chamber is ~1 Torr; temperatures were hold at 130 °C, 530 °C, and 930 °C for the S-, MoO$_3$-source and substrate; and the growth process lasts usually about 50 min. Supplementary Fig. 2 demonstrates an optical image of high-quality 4-inch ML-MoS$_2$ wafer with large domain sizes between 200 μm and 500 μm. These domains are well-stitched together to form a continuous film with 100% coverage.

### Deposition of High-κ dielectric layers

ALD of HfO$_2$ on Ti/Au/Ti (1 nm/5 nm/1 nm) local metal gates were carried out by Savannah-100 system (Cambridge NanoTech. Inc.) with H$_2$O and tetrakis dimethylamino hafnium (TDMAH) as precursors. Prior to the deposition process, we used O$_2$ plasma to treat the Ti/Au/Ti electrode surface by reactive ion etching (RIE) to oxidize the surface Ti layer. The oxidized layer (TiO$_{2-x}$) acts as a buffer layer for dense high-κ dielectrics deposition. During the deposition, 20 sccm high-purity nitrogen was flowed as the carrier and purge gas; the reactor pressure was ~3 Torr; and the TDMAH precursor was heated to 75 °C. The deposition temperature is 110 °C or 150 °C for flexible samples, and 200 °C for rigid samples. The pulse and reaction time was 0.015/0.15 and 60/60 s for TDMAH/H$_2$O precursors with deposition rate around 1 Å per cycle.

### Device fabrications and measurements

TFTs and logic devices were fabricated via standard microfabrication processes such as e-beam lithography or UV lithography, oxygen RIE, e-beam evaporation and lifting-off. Transfer of ML-MoS$_2$ films from sapphire to target substrates were assisted by wet etching in KOH solution (1 Mol/L, 110 °C). The process is illustrated in Supplementary Fig. 3. Au contact electrodes (10–15 nm) with clean interfaces were fabricated by e-beam evaporation with an ultra-low deposition rate of ~0.01 Å·s$^{-1}$. All the electrical measurements were carried out in a Janis probe station at a base pressure of 10$^{-6}$ Torr with an Agilent semiconductor parameter analyzer (B1500, high resolution modules) and Agilent digital oscilloscope (DSO-X 3054 A) at room temperature.

### STEM characterizations

The cross-sectional STEM characterizations were carried out by an aberration corrected JEOL ARM 300 F transmission electron microscope operated at 300 kV.

## Data availability

The Source Data underlying the figures of this study are available at https://doi.org/10.6084/m9.figshare.22776989. All raw data generated during the current study are available from the corresponding authors upon request.

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

## Acknowledgements

This work was supported by the National Key Research and Development Program (Grant No. 2021YFA1202900, 2021YFA1400502), the Strategic Priority Research Program of Chinese Academy of Sciences (CAS, grant No. XDB30000000), the National Science Foundation of China (NSFC, grant No. 61888102, 11834017, 61734001, and 62122084), and the Key-Area Research and Development Program of Guangdong Province (Grant No. 2020B0101340001). The authors also thank Prof. Aizi Jin, Prof. Tianzhong Yang and Dr. Xiao Guo in IOP-CAS for the assistance on short channel device fabrications and CS-STEM sample preparations.

## Author contributions

J.Ta. fabricated the devices and carried out the electrical measurements with the help of J.Ti., N.L., Y.P., Xiu.L., Y.Z., C.H., B.H., and Y.C. J.Ta., J.Ti.,

S.W., J.L., Y.G., and Da.S. performed the high-κ dielectric deposition. Q.W. performed the growth of ML-MoS$_2$ films. Xia.L. J.Ta., Y.J. and X.B. prepared the TEM samples and performed the TEM characterizations. G.Z., Do.S., W.Y., R.Y., and L.D. supervised the project. J.Ta. and G.Z. analyzed data and wrote the manuscript. All authors discussed and commented on the manuscript.

## Competing interests

The authors declare no competing interests.
