## [Peer Review File · Nature Communications]

Low power flexible monolayer MoS₂ integrated circuitsEditorial Note: This manuscript has been previously reviewed at another journal that is not operating a transparent peer review scheme. This document only contains reviewer comments and rebuttal letters for versions considered at *Nature Communications*.

Point to point response letter

Problem 1. Inverter Simulation

In this simulation, the author used the level 62 Poly-Si model in HSPICE platform to realize the MoS₂ transistor. However, this simulation has overestimated the overall performance of the inverter by improperly utilizing the merit of device scaling and advanced Si manufacturing technologies even if they just used general lab equipment, and many CMOS processes do not applicable to MoS₂. Especially, we could confirm that the simulation result was exaggerated by unsuitable zero contact resistance and short channel effects such as DIBL. For instance, the author used the value of AT, BT parameters which indicate how the DIBL effect is strong in this Poly-Si model as the model default value. Because of this mistake, it always produces excellent inverter performance even if arbitrary transistor characteristics such as mobility (μ) of 0.1 cm²/Vs are used. Therefore, the excellent inverter performance was not because of their MoS₂ characteristics but because of certain parameters related to advanced Si fabrication technique and are intrinsically included in this Poly-Si model. In addition, it has been reported that the DIBL effect for a simple configuration of MoS₂ transistor is higher than 100 mV/V if special structure design such as double gate or FinFet structure is not included (Figure 2). However, the estimated DIBL value in author's simulation are about 10 mV/V which is far below than previous literatures since they just used model default value. To confirm the DIBL effect and find better condition, we tried several different AT parameters values based on literatures and unfortunately, the inverter performance was not high as they claimed. Especially, the gain values were not comparable to both simulation (~ 257) and the device result (~397) which claimed in this paper (Figure3).

Author response:

In response to the referee's previous arguments on "**if high-gain inverters and ring oscillators can work**", we have carried out additional simulations using the level 62-Poly-Si model in HSPICE platform. As shown in Fig. R1, our simulation indeed confirm that the inverters can work based on the experimental output results of MoS₂ FET. However, once again the reviewer is unsatisfied and biased through their additional unreasonable simulation results. This comment is far beyond the scope of our manuscript, as the main purpose of our work to demonstrate experimental results. If the simulation is inconsistent with our experimental results, it indicates that the simulation needs to be improved. The way of criticizing the solid experiments due to the unreasonable parameter setting in the simulations is aggressive, and thus we don't agree with their comments.

Although the simulations cannot fully reproduce our experimental results (due to some parameters used in simulations are uncertain), our simulations on the MoS₂ FETs can demonstrate the working mechanism of MoS₂ inverter and ring oscillator. Experimentally, we construct the inverter and ring oscillator based on long channel FETs, so we don't have short channel effect, such as DIBL effect, etc. Thus, during the simulation, we just adopt the parameters of Si FET without considering the contact resistance or DIBL effect. The simulated output and transfer curves (solid line) could fit the experiment results (dash circle) well, which includes the high output resistance and SS, etc. (Fig. R1). Our simulation suggests that the inverter could work with high inverter gain.

From the experimental point of view, the high inverter gain is realized by optimizing the output performance of MoS₂ FET units: (1) introducing the thin HfO₂ layer (10 nm) as dielectric layer, which could confirm the strong gate controllability of the FET unit with sharp SS and high output resistance; (2) use the high-quality monolayer MoS₂ film with large domain size, which could ensure the high output current densities, on/off ratio; (3) optimize the contact quality with small contact resistance. The above optimizations could ensure the inverter work with high performance. These issues have been all included in our manuscript as highlighted in red.

Fig. R1. The simulation of NMOS MoS₂ inverter implemented with the 62-level SPICE model calibrated with the measured electrical properties of MoS₂ FET units with channel length/width of 10/40 μm. (a-b) Measured and simulated output curves and transfer curves of MoS₂ FET. (c) Measured and simulated voltage transfer curves (VTCs) of inverters. (d) The corresponding inverter gains. (Dash circle, experimental data; Solid line, simulated data.)

Problem 2. V_{th} Selection

Furthermore, the author claimed that the excellent inverter performance was obtained by designing the load transistor to be depletion-mode ($V_{th} < 0$) and the driver transistor to be the enhance mode ($V_{th} > 0$) contains serious logical errors considering the V_{th} of individual transistors cannot be controlled by this manner. For example, in previous literature, depletion mode and enhance-mode were established by varying the work function of gate metal (Figure 4) or introducing a double gate structure (Figure 5). However, the author's used the same gate metal with a single gate structure which is not able to precisely control the V_{th} of each transistor. Therefore, the author

should clearly explain how the depletion mode and enhance mode of transistors could be selected based on the device structure shown in main figure 3b and the V_{th} distributions shown in main figure 2c. Furthermore, the depletion-mode and enhance-mode design have already been addressed in previous simulations we present. However, no evidence was found for the inverter performance to be reached as claimed in this article, especially gain of 397 at $V_{dd} = 1V$, 1000 at $V_{dd} = 2V$. The author did not provide any reasonable explanation for this result. We also double checked the inverter and depletion, enhance mode design in the level 6 IDS MOSFET SPICE model (Figure 6). With an TFT model having very similar I-V characteristic with the author's MoS_2 device, the inverter gain was still not comparable to the author's result, and rather showed similar behavior to the literatures (Ref 1, Ref 2).

Author response:

We agree with Reviewer #1 that the optimizing the threshold voltage (V_{th}) could make the inverter work much better, and we will solve this concern in our future work. In present experiment, the inverter could work well based on our device configuration design.

Experimentally, we have tested a great quantity of devices, and a large proportion of MoS_2 inverters could work with excellent output performances, especially for the inverter gain. We demonstrated the most typical one in our manuscript as shown in Fig. 3c. The experiment result is the solid proof to demonstrate the working performance of the inverters.

We carried out further simulation to demonstrate the results with the request from reviewer 1 in the last reviewing process. For our simulation, we construct the inverter that the load and drive MoS_2 transistors have 0.5V threshold voltage mismatch which is used for simulating the experimental results and demonstrate that our experimental results can be repeated under such parameter settings. When the load transistor work in the depletion-mode and the driver transistor work in the enhance mode, the inverter could have comparable output performance with our experiment results.

During the simulation, it's easy for reviewer to tune the setting parameters and make the device work with poor performance or fail to work, but it doesn't mean our devices can not work. The reviewer's comments are unreasonable and fragile.

Problem 3. Inconsistency with Previous Literatures

Also, the author's experimental and simulation results were very inconsistent with previous literature conducted with similar MoS_2 transistors and enhance, depletion mode design. For instance, the gain of inverter demonstrated by a single crystalline MoS_2 which has the mobility of about $300 \text{ cm}^2/Vs$, 88 mV/decade subthreshold swing, 10^7 on/off ratio shows only about 5 at $V_{dd} = 1V$ (ref 1). Another study using double gate structure MoS_2 TFT that has $18 \text{ cm}^2/Vs$ mobility, 60 mV/decade subthreshold swing, larger than 10^7 on/off ratio also has reported the gain value of 3.5 ~ 5.3 at $V_{dd} = 1V$ (ref 2). Thus, our simulation results, in which inverter gain values has appeared as 0.5 ~ 2 at $V_{dd} = 1V$ are better matched with other reported values while the author has not been to clearly explain this inconsistency.

Ref 1. Wang, Han, et al. "Integrated circuits based on bilayer MoS_2 transistors." *Nano letters* 12.9 (2012): 4674-4680.

Ref 2. Liao, Fuyou, et al. "High-performance logic and memory devices based on a dual gated MoS₂ architecture." *ACS Applied Electronic Materials* 2.1 (2019): 111-119.

Author response:

In reference 1 (*Nano letters* 12.9 (2012): 4674-4680), authors adopt exfoliated bilayer MoS₂ as channel materials for demonstrating the N-type inverter. They use Pd and Al as gate electrodes for constructing load and drive FET of the inverter. Owing that these metals have different work function, the FETs could be work at the enhanced- and depletion-mode. However, they didn't optimize the output performance of the inverter based on the exfoliated flakes and only demonstrate the inverter function. What's more, as shown in Fig. R2, the output resistance is on the order of 0.7 MΩ for the bilayer MoS₂ inverter in Ref. 1 work (*Nano letters* 12(9), 4674-4680, 2012); however, the output resistance is on the order of >100 MΩ as shown in in our work of Extended data Fig. 8b. Based on the results as shown in Extended data Fig. 8, the intrinsic gain (g_m/g_d) could over than 1000. This analysis is consistent with our results of the inverter with high gain.

In reference 2 (*ACS Applied Electronic Materials* 2(1) 111-119, 2019), authors adopt dual-gate MoS₂ FET as the drive FET unit and a 100 MΩ resistor as load FET as shown in Fig. R3, this design is totally different with our work. They also don't focus on the inverter performance in this work. Their results focus on that the inverter output could improve when the drive FET working in dual-gate mode with stronger gate-controllability. Also, the trip point also shifts negative when changing the threshold voltage of the drive FET, as the V_{th} would shift under different VTG values.

[redacted]

Fig. R2. The comparisons of the output performance of the MoS₂ FETs. (a) The output behavior of the bilayer MoS₂ FET in Ref. 1 (*Nano letters* 12(9), 4674-4680, 2012). (b) The output behavior of the MoS₂ FET in our work (Extended Data Fig. 8b).

[redacted]

Fig. R3. The output of DG-MoS₂ inverter from the reference 2 (*ACS Applied Electronic Materials* 2(1) 111-119, 2019). (a) The output behavior of MoS₂ inverter work with single back-gate (BG) and dual-gate (DG) mode. (b) The output results of DG-MoS₂ inverter working under different VTG.

Problem 4. Ring Oscillator

We arise serious concerns about the statistical simulation that the author provides. The actual distribution functions in the simulation code are different from what they mentioned, and it is also not matched with their device result in main figure 2C. In the simulation code, the V_{th1} and V_{th2}

were set to be centered at -0.22V and 0.35V with a standard deviation of 0.2V while the author claimed this simulation was carried out with V_{th} of -0.04V on centered which consequence of the ring oscillator can oscillate and the yield becomes much higher (Figure 7). We could confirm that the ring oscillator didn't operate with an exact distribution function setting of centered at -0.04 V and standard deviation of 0.2 V (Figure 8).

Author response:

Our simulated results were used for verifying our experimental ring oscillator results. In our simulation code, we selected the V_{th1} and V_{th2} centered at -0.22V and 0.35V with a standard deviation of 0.2V for ensuring large noise margin for every inverter stage (here V_{th1} and V_{th2} also locate within the experimental FET threshold voltage distribution range as shown in Figure 2c), which is key factor for the oscillation of ring oscillator. If the referee set the parameters which cannot make the ring oscillator start to work, that's meaningless. As shown in Figure 4, and the supplementary video, we have demonstrated and supplied reliable experimental data about the stable outputs of ring oscillators. The simulation results requested by the referee is only used for demonstrating that the ring oscillator can work reliable. What's more, the simulation could only guide our experiments. Based on our solid experimental results, our ring oscillator could work for sure. Besides, we have also fabricated flexible ring oscillators using much thicker oxides (to lower the oscillating frequency for artificial retina applications) and these oscillators also work very well, the results have been published recently in the journal of *ACS Nano* (refer to: *ACS Nano* 2023, <https://doi.org/10.1021/acsnano.2c06921>).

REVIEWER COMMENTS

Reviewer #5 (Remarks to the Author):

The authors report on wafer-scale rigid and flexible MoS₂ electronics with high performance. In terms of high device performance, this mostly refers to rigid transistors with high on-currents similar to best previous reports. Flexible device performance is not clearly presented, but extracted mobility on flexible devices suggests high quality and potentially high device performance. The gate dielectric quality on rigid substrates is impressive showing remarkable stability to fields of 1V/nm. A comparable analysis on flexible substrates is however missing. The low-power argument the authors provide for their flexible inverters might be true compared to other unipolar flexible devices, but not clearly benchmarked and certainly not to be claimed in a general non-comparative setting. Lastly, ring oscillators seem to provide a decent improvement compared to previous flexible MoS₂ literature in terms of operating voltages. Overall, the manuscript has potential, but the reviewer advises that the authors provide extensive additional data and very clear benchmarking and comparisons with the respective rigid or flexible (not both intermixed) device literature. Please find more specific technical comments below:

1) In the abstract, authors mention several aspects of high performance that they claim to achieve for their flexible transistors. Some of these properties are more fitting for low-power than for high-performance. Furthermore, high drive current is only demonstrated for rigid transistors and not for flexible ones. Thus, the way the abstract is written is misleading. The authors need to show high-performance metrics on flexible devices to put such claims into the abstract. Otherwise please refrain from the "high-performance" statement.

2) The authors provide strong claims for highest grain size to date (L71-31) and quality (L79-80 "Such high-quality materials offer the best electronic quality and device performance ever achieved,"). These need to be fully supported by results and benchmarking. A detailed benchmark comparison which includes grain size and electronic quality is missing. That these claims are not overreaching is at least doubtful also given some recent developments in the field (see e.g., <https://doi.org/10.1038/s41586-022-05524-0>). Furthermore, as already mentioned above, it is unclear if these claims are in relation to flexible devices or rigid devices. The authors need to clearly distinguish literature and their own results in these two domains and report and compare each separately (in Table S2 it is all mixed providing a wrong picture on the performance of flexible devices, Fig 2e shows rigid device data despite being part of a figure on flexible device characteristics).

3) The authors use different HfO₂ dielectric thicknesses for the rigid and flexible substrates (5 nm vs. 10 nm). What is the reason for the higher thickness needed for flexible devices? Is it the roughness of the PET substrate (please measure it)? The smooth dielectric and high breakdown quality are shown in Extended Data Fig. 2 only for the 5 nm HfO₂ on rigid substrates. The authors also need to show such an analysis for flexible devices with their respective HfO₂ material.

4) In relation to my previous comment, the authors need to analyze the gate leakage current in more detail. No gate currents are shown for flexible devices. Is the gate leakage higher in such devices? The data should be shown and discussed. Furthermore, please comment on the noise limit for current measurements, as it seems that Fig 1g (rigid devices) show off-state drain current and gate current in the noise level.

5) How can the authors normalize Fig 1e by channel length without taking into account contact resistance? A channel length normalization seems meaningless otherwise.

6) Why is the mobility different for different dielectrics and reduced for lower EOT as stated in L144-147?

7) The authors claim low contact resistance of 0.59 k Ω · μm . Based on an extraction with channel lengths of 5-75 μm (Fig 2d). Usually, without going to small channel lengths in the sub-micron regime, the errorbars on the contact resistance extraction are very high. What is the fitting error that the authors obtain? And thus, how confident are they about this value?

8) The argument on having a better device architecture due to lower water permeability in L211-219 while using uncapped devices is meaningless. I advise to remove this discussion if there is no further evidence of improved resistance against water permeability in capped devices.

9) Bending and strain: Authors state that the maximum strain is 2.5% which is very high and it is doubtful that the dielectric and metals are not cracking at those strain values. Did the authors verify that strain is imparted correctly onto the device? Please provide Raman measurements of MoS₂ under strain to confirm that 2.5% is really encountered by the devices. Please also report on the gate leakage current upon straining because a cracked dielectric could lead to increased gate leakage.

10) As stated in L234, how do authors suggest to "adopting CMOS units as building blocks" in a purely n-type unipolar technology?

11) Fig 3c: The switching transition is below $V_{dd}/2$. As the authors write in L226-228: "The trip point of the inverter is when V_{out} equals V_{in} and of particular importance for low-power electronics as it determines the driving voltage (V_{dd}) of integrated logic circuits." The noise margin should be extracted in a measurement where V_{in} is swept up to the value of V_{dd} . Then it becomes obvious that the noise margins for high and low will be asymmetric. It is misleading to extract this with a V_{in} range not up to V_{dd} . As the authors show in Fig 3e, they operate the logic gates at $V_{in} = V_{dd} = 0.5 V$

12) How is this technology low power or energy efficient if a unipolar technology is used and a significant current ($I_{dd} > 0.4 nA$) flows at high V_{in} (Extended Data Fig 8f). Additionally, regarding the low power arguments: A threshold voltage averaging around 0V is not desirable, because the device should be fully off at $V_{GS} = 0V$, requiring a positive V_T , good SS and low off current.

13) Extended Data Fig 3b: At which frequency was the MOSFET capacitance measured and what was the active area assumed? Due to channel accumulation the whole MoS₂ will be an active part of the parallel plate capacitor in accumulation and due to a parasitic MoS₂ resistance there could be an underestimated capacitance if measured at high frequencies. Did the authors see a pronounced frequency dispersion in the MOSFET-capacitors?

14) Ring oscillator frequency: The authors show good results in the ns-regime. To evaluate this result, they should clearly state the device dimensions and gate-to-drain/source overlaps. Their statement in line 256-257 "Note that the channel length/width of each TFT unit varies from 1 μm to 40 μm for the ROs." is unclear in this regard. In addition, I cannot find Supplementary Figure S3. Did the authors mean Figure S2 instead? Figure S2 is not sufficiently described to understand what kind of parasitic optimization was done. Please provide more details.

15) Ring oscillator benchmarking Fig 4f. The authors have missed crucial work on CNTs from literature which actually outperforms their circuits (<https://doi.org/10.1038/s41467-022-34621-x>). Please include this in the benchmark. Nevertheless, the here presented result seems to be a good improvement over previous flexible MoS₂ literature mostly in regards to operating voltage, as the authors have already achieved similar stage delays in their own prior work (<https://doi.org/10.1038/s41928-020-00475-8>).

Point-to-point response letter

The authors report on wafer-scale rigid and flexible MoS₂ electronics with high performance. In terms of high device performance, this mostly refers to rigid transistors with high on-currents similar to best previous reports. Flexible device performance is not clearly presented, but extracted mobility on flexible devices suggests high quality and potentially high device performance. The gate dielectric quality on rigid substrates is impressive showing remarkable stability to fields of 1V/nm. A comparable analysis on flexible substrates is however missing. The low-power argument the authors provide for their flexible inverters might be true compared to other unipolar flexible devices, but not clearly benchmarked and certainly not to be claimed in a general non-comparative setting. Lastly, ring oscillators seem to provide a decent improvement compared to previous flexible MoS₂ literature in terms of operating voltages. Overall, the manuscript has potential, but the reviewer advises that the authors provide extensive additional data and very clear benchmarking and comparisons with the respective rigid or flexible (not both intermixed) device literature. Please find more specific technical comments below:

Author response:

We sincerely appreciate the referee's efforts and time in reviewing our manuscript and positive comments here. The constructive comments and suggestions are quite helpful in improving the quality of our work.

In the revised manuscript, we include further discussions and analysis of the dielectric quality of 5 nm HfO₂ deposited on flexible substrate and measured the corresponding device properties. Further, we also separate comparisons between flexible and rigid substrates to compare detailed device parameters with previous representative literature works related to MoS₂ flexible electronics (Supporting Information Table S1-S4), power consumption comparisons with representative ultra-low power transistor works (Supporting Information Table S6), and further revise the main text/figures as well. The technical comments have been addressed properly in the revised manuscript as highlighted in red. Please see the detailed responses below.

1) In the abstract, authors mention several aspects of high performance that they claim to achieve for their flexible transistors. Some of these properties are more fitting for low-power than for high-performance. Furthermore, high drive current is only demonstrated for rigid transistors and not for flexible ones. Thus, the way the abstract is written is misleading. The authors need to show high-performance metrics on flexible devices to put such claims into the abstract. Otherwise please refrain from the "high-performance" statement.

Author response:

Thank you for the comments here. We agree that the low-power theme is more suitable for this work.

Per the referee's comments, we have revised our manuscript as highlighted in red and summarized below:

(1) Title: '**Low power flexible monolayer MoS₂ integrated circuits**'

(2) Abstract: '*Based on high-quality wafer scale ML-MoS₂, in this work we develop ultra-thin high-κ/metal gate technique for thin film transistors on both rigid and flexible substrates. The rigid devices can*

be operated in the deep-subthreshold regime with low power while having high performances, such as negligible hysteresis, sharp subthreshold slope, high current density, and ultra-low leakage current. We further realize large-scale flexible ICs. At low operation voltages below 1 V, fully functional flexible ICs were successfully demonstrated. Our process could move forward a key step towards using energy-efficient flexible ML-MoS₂ ICs in portable, wearable, and implantable electronics.'

(3) Introduction: *'In this work, we demonstrate the realization of ~~high performance and low-power flexible ICs based on ML-MoS₂~~ 'As-grown films are strictly monolayer, uniform across the entire substrates, and have large domain sizes over 200 μm (refer to the Extended Data Fig. 1 and Supporting Information Table S1), ~~the largest over previously reported epitaxial ML-MoS₂ films~~ 'Such high-quality materials offer ~~the best electronic quality and device performance ever achieved, as will be demonstrated in the following device characterizations~~ a desirable choice for realizing large-scale flexible ICs. ML-MoS₂ Here, the flexible ICs are fabricated via a gate-first technology in which ultra-thin high-κ dielectrics are deposited on gate electrodes to achieve a record-low effective oxide thickness (EOT). Due to the enhanced gating efficiency, our ML-MoS₂ flexible TFTs can work under operation voltages below 1 V with ~~state-of-the-art performances comparable to, or even better than, their rigid counterparts~~'.*

2) The authors provide strong claims for highest grain size to date (L71-31) and quality (L79-80 "Such high-quality materials offer the best electronic quality and device performance ever achieved,"). These need to be fully supported by results and benchmarking. A detailed benchmark comparison which includes grain size and electronic quality is missing. That these claims are not overreaching is at least doubtful also given some recent developments in the field (see e.g., <https://doi.org/10.1038/s41586-022-05524-0>). Furthermore, as already mentioned above, it is unclear if these claims are in relation to flexible devices or rigid devices. The authors need to clearly distinguish literature and their own results in these two domains and report and compare each separately (in Table S2 it is all mixed providing a wrong picture on the performance of flexible devices, Fig 2e shows rigid device data despite being part of a figure on flexible device characteristics).

Author response:

Sorry for the misleading description *'the best electronic quality and device performance ever achieved'* in our manuscript without solid proof and mixing information. For clarifying this disagreement here, we make a comparison with some typical literature works as shown below in Table R1. We will start from two aspects for comparisons:

(1) In order to gain better MoS₂ film quality for large-scale electronic applications, the most important challenge is reducing the density of domain boundaries. Researchers usually adopt the following two solutions, such as increasing the domain size or controlling the domain orientation during the growth.

Compared with the domain size column in Table R1, (a) the works of ref. 4 and ref. 6 are mainly focused on increasing the domain size. For ref. 4, the largest domain size is 400 μm with multi-orientation, while the device mobility is still not very high. For ref. 6 (our recent work), the domain size is ~180 μm with much higher device mobility of ~70 cm²·V⁻¹·s⁻¹ on rigid substrate. (b) ref 5, ref 8 and ref 10 are mainly focused on realizing the single-domain MoS₂ film growth, and corresponding device mobility could be enhanced to 60-80 cm²·V⁻¹·s⁻¹. In this work, we still focus on increasing the domain size to reduce the domain boundary density, and 4-inch wafer-scale MoS₂ film with 200-500 μm domain size are realized with great effort. Most importantly, we fabricated the devices with pretty high mobility of ~70 cm²·V⁻¹·s⁻¹, which is comparable with the single domain films, because the device size is usually less than the domain size.

(2) Another important opportunity for 2D semiconducting materials is their flexibility nature as compared with traditional rigid semiconducting materials. Thus, we focus on the flexible MoS₂ electronics with good performance. As shown below, ref. 7 and ref. 9 are mainly focused on this topic. In ref. 7 (our recent work), the main restriction for wearable application is the pretty high operation voltage (20-70 V) based on the Al₂O₃ dielectric layer with thickness of 30-80 nm. In ref. 9, researcher realize that lower the operation voltage to 15 V, but with lower device mobility. In this work, we implement the flexible MoS₂ thin film transistors with much smaller operation voltages (within 3-7 V), while still preserving high device mobility, and the logic function circuits could be operated reliable under 1 V supply voltages. More comparative details could refer to Supporting Information Table S3.

We appreciated the reviewer's comments and we have separated the comparison of benchmark parameters regarding flexible and rigid devices in our revised manuscript, and revised Table S2 and Fig. 2e correspondingly for better demonstration. Also, we have revised the declaration as '*Such high-quality materials offer the best electronic quality and device performance ever achieved, as will be demonstrated in the following device characterizations a desirable choice for realizing large-scale flexible ICs.*' in our revised manuscript as highlighted in red. Besides, we also add the comparisons of MoS₂ growth in Supporting Information as Table S1.

Table R1. The comparison of the MoS₂ growth and device benchmarks.

Reference	Year	Domain size	Orientation	Device mobility	Operation voltage	Substrate
1. Polycrystalline MoS ₂ on SiO ₂	2014	<1 μm	>2	7 cm ² ·V ⁻¹ ·s ⁻¹	60 V	Rigid
2. Polycrystalline MoS ₂ on fused silica	2015	<1 μm	>2	30 cm ² ·V ⁻¹ ·s ⁻¹	>30 V	Rigid
3. MoS ₂ epitaxy on sapphire	2017	2-5 μm	2	40 cm ² ·V ⁻¹ ·s ⁻¹	60 V	Rigid
4. MoS ₂ on glass	2018	400 μm	>2	11 cm ² ·V ⁻¹ ·s ⁻¹	30 V	rigid
5. MoS ₂ epitaxy on Au	2020	Single domain	1	10 cm ² ·V ⁻¹ ·s ⁻¹	20 V	rigid
6. MoS ₂ epitaxy on sapphire (our work)	2020	180 μm	2	70 cm ² ·V ⁻¹ ·s ⁻¹	100 V	rigid
7. MoS ₂ epitaxy on sapphire (our work)	2020	20 μm	2	55 cm ² ·V ⁻¹ ·s ⁻¹	20-70V	flexible
8. MoS ₂ on sapphire	2021	Single domain	1	80 cm ² ·V ⁻¹ ·s ⁻¹	10 V	rigid
9. MoS ₂ on SiO ₂	2021	~20 μm	/	27 cm ² ·V ⁻¹ ·s ⁻¹	15V	flexible
10. MoS ₂ on HfO ₂	2023	Single domain	1	62 cm ² ·V ⁻¹ ·s ⁻¹	30 V	rigid
This work	/	200-500μm	2	70 cm ² ·V ⁻¹ ·s ⁻¹	7V	flexible

References:

1. Zhang, Jing, et al. "Scalable growth of high-quality polycrystalline MoS₂ monolayers on SiO₂ with tunable grain sizes." ACS nano 8.6 (2014): 6024-6030.
2. Kang, Kibum, et al. "High-mobility three-atom-thick semiconducting films with wafer-scale homogeneity." Nature 520.7549 (2015): 656-660.
3. Yu, Hua, et al. "Wafer-scale growth and transfer of highly-oriented monolayer MoS₂ continuous films." ACS nano 11.12 (2017): 12001-12007.
4. Yang, Pengfei, et al. "Batch production of 6-inch uniform monolayer molybdenum disulfide catalyzed by sodium in glass." Nature communications 9.1 (2018): 979.
5. Yang, Pengfei, et al. "Epitaxial growth of centimeter-scale single-crystal MoS₂ monolayer on Au (111)." ACS nano 14.4 (2020): 5036-5045.
6. Wang, Qinqin, et al. "Wafer-scale highly oriented monolayer MoS₂ with large domain sizes." Nano Letters 20.10 (2020): 7193-7199.
7. Li, Na, et al. "Large-scale flexible and transparent electronics based on monolayer molybdenum disulfide field-effect transistors." Nature Electronics 3.11 (2020): 711-717.
8. Li, Taotao, et al. "Epitaxial growth of wafer-scale molybdenum disulfide semiconductor single crystals on sapphire." Nature Nanotechnology 16.11 (2021): 1201-1207.
9. Daus, Alwin, et al. "High-performance flexible nanoscale transistors based on transition metal dichalcogenides." Nature Electronics 4.7 (2021): 495-501.
10. Kim, Ki Seok, et al. "Non-epitaxial single-crystal 2D material growth by geometric confinement." Nature (2023): 1-7.

3) The authors use different HfO₂ dielectric thicknesses for the rigid and flexible substrates (5 nm vs. 10 nm). What is the reason for the higher thickness needed for flexible devices? Is it the roughness of the PET substrate (please measure it)? The smooth dielectric and high breakdown quality are shown in Extended Data Fig. 2 only for the 5 nm HfO₂ on rigid substrates. The authors also need to show such an analysis for flexible devices with their respective HfO₂ material.

Author response:

Thanks the reviewer for these comments. We also implemented the deposition of 5 nm-HfO₂ dielectric layer on flexible substrate and measured the device properties as shown in Fig. R1. In Fig. R1b, the leakage level (I_g) is around the level from 1 pA to 10 nA with the gate (V_g) range from -0.5V to 4V, which is much higher than leakage current level of short channel devices fabricated on SiO₂ substrate as shown in Extended Data Fig. 2c.

The ultra-thin high- κ deposition technique of 5-nm HfO₂ also works on flexible substrate, as shown in Fig. R1a. The transfer curves of the 5-nm HfO₂ devices still show good uniformity with on/off ratio $\sim 10^6$ - 10^7 , $I_{SS} \sim 70$ mV/dec, $I_{off} \sim 100$ pA. The subthreshold swing properties is much better than 10 nm-HfO₂ as shown in Fig. 2 in our manuscript, however, the main issue is the device yield of 5 nm-HfO₂ on flexible substrate, which is around 60% due to the leakage issue (calculated from 100 FETs) and much lower than $\sim 96\%$ (calculated from 500 FETs) of 10 nm-HfO₂ on flexible substrate. The device yield of the FET units is very important for realizing the integrated circuit. Thus, to ensure the successful rate of the integrated

logic circuits in the following experiments, we adopt 10 nm HfO₂ as dielectric on flexible substrate for realizing the low power electronic application. We have added such data based on 5-nm HfO₂ to our revised manuscript as Fig. S4 in Supporting Information.

The surface roughness of the PET substrate used in our experiment is 0.72 ± 0.1 nm as shown in Fig. R2, which is higher than the SiO₂ surface roughness of ~ 0.2 nm. The higher leakage of the 5 nm-HfO₂ on PET substrate may come from the following aspects: (1) The increased surface roughness of the PET substrate; (2) The surface dangling bond or polymer residues of the PET substrate, which may lead to poor dielectric properties of the HfO₂ layer.

Fig. R1. The electrical properties of flexible MoS₂ FETs based on 5-nm HfO₂ as dielectric layer. (a) The transfer curves. (b) The corresponding statistics of leakage current.

Fig. R2. The AFM image of the PET substrate with surface roughness of $\sim 0.72 \pm 0.1$ nm.

4) In relation to my previous comment, the authors need to analyze the gate leakage current in more detail. No gate currents are shown for flexible devices. Is the gate leakage higher in such devices? The data should be shown and discussed. Furthermore, please comment on the noise limit for current measurements, as it seems that Fig 1g (rigid devices) show off-state drain current and gate current in the noise level.

Author response:

Thanks for the reviewer's comments. The leakage current of the flexible FETs fabricated on 5 nm and 10 nm HfO₂ as shown in Fig. R3. We could see that the leakage current (I_g) level of 10-nm HfO₂ on PET substrate is on the level from 0.1 pA to 1 nA with the gate (V_g) range from -1 V to 7 V, which is 1-2

orders less than the leakage level of the devices fabricated on 5-nm HfO₂ on PET substrate. We have added the leakage current analysis to Supporting Information as Fig. S4. Also, we have added the leakage current and device yield discussion in the revised manuscript as ‘Note that we also realize the fabrications of ML-MoS₂ FETs based on 5-nm HfO₂ as shown in Supporting Information Fig. S4. Considering the device yield is ~60% and higher leakage current level, we mainly adopt 10 nm HfO₂ as dielectric layer for integrated logic gates in following experiments.’ on page 5, para 2 as highlight in red.

Also note that we use an Agilent semiconductor parameter analyzer (B1500, high resolution modules) for electrical measurements and the noise level is between the range of 1E-14 to 1E-12 A.

Fig. R3. The leakage current of MoS₂ FETs fabricated on flexible PET substrate. (a) 5-nm HfO₂. (b) 10-nm HfO₂.

5) How can the authors normalize Fig 1e by channel length without taking into account contact resistance? A channel length normalization seems meaningless otherwise.

Author response:

Thank you for the comments. In Fig. 1e, we use the normalized current to compare the working range of the gate voltage (V_g) based on different dielectric thickness. Note that in such comparison, we only want to compare the scaling effect of gate voltage based on different dielectric layer thickness, such as 5-nm, 10-nm HfO₂ and 30-nm Al₂O₃, thus we adopt the devices with the same channel length $L_{ch}=5 \mu\text{m}$ and channel width of $40 \mu\text{m}$ and normalize the current by channel length and channel width for better comparison. We could see that the gate voltage reduces from 15 V to 3 V with the scaling down of dielectric thickness to 5 nm.

However, the current densities would be influenced by the contact resistance, especially for the short channel devices. In this comparison, the contact resistance is roughly on the same level based on the same device structure and contact fabrication techniques.

To avoid the misleading, we add device parameters in figure caption of Fig. 1 in the revised manuscript ‘e. Normalized transfer curves of MoS₂ TFTs ($L=5 \mu\text{m}$, $W=40 \mu\text{m}$) at $V_{ds}=1 \text{ V}$ with 5 nm, 10 nm HfO₂ and 30 nm Al₂O₃ as the dielectric layer, individually.’ as highlighted in red.

6) Why is the mobility different for different dielectrics and reduced for lower EOT as stated in L144-147?

Author response:

Thank you for the reviewer's comments. These discussions are upon the the request from the previous reviewer who suggested that we discuss the EOT effect on the device mobilities. We thus calculated the device mobility based on the transfer curve of FETs in Fig. 1e with the 5-nm HfO₂, 10 nm HfO₂, and 30-nm Al₂O₃ as dielectric respectively and make a comparison between them. The results indicate that 10-nm HfO₂ has the best device mobility value as comparing with the 5-nm HfO₂ and 30-nm Al₂O₃, which may come from the following aspects:

(1) We only calculated one single device based on each device type. The device-to-device mobility variation is large and cannot be ignored, and it may induce the difference between them. To be cautious, we compare a quality of FET results as shown in Fig. R4. Based on the calculated device mobility distribution, the average device mobility of 5-nm HfO₂ is $55 \pm 13 \text{ cm}^2 \cdot \text{V}^{-1} \cdot \text{s}^{-1}$, which is slightly smaller than 10-nm HfO₂ of $68 \pm 19 \text{ cm}^2 \cdot \text{V}^{-1} \cdot \text{s}^{-1}$ and 30-nm Al₂O₃ of $65 \pm 20 \text{ cm}^2 \cdot \text{V}^{-1} \cdot \text{s}^{-1}$.

(2) Based on the mobility results, we could observe that the device mobility of 5 nm HfO₂ is slightly smaller than 10 nm HfO₂ FETs. It may be due to the dielectric quality of 5-nm HfO₂ not being as good as 10-nm HfO₂ and inducing slightly poor output performance. Moreover, the device mobility of 10-nm HfO₂ FETs is comparable with the 30 nm Al₂O₃ FETs, which indicates the gate voltage scaling effects originating from the dielectric with thinner thickness and higher dielectric constant.

(3) Even though the device mobility of 5 nm HfO₂ is slightly smaller than the other two types, we could get a tentative conclusion ‘with decreased t_{HfO_2} and increased ϵ_r , the gate voltages (V_g) can be reduced from 15 V to 3 V and the SS can be reduced from $250 \text{ mV} \cdot \text{dec}^{-1}$ to $75 \text{ mV} \cdot \text{dec}^{-1}$, while preserving the on-state current (I_{on}) densities and on/off ratios.’ in our manuscript.

For reduce the misleading information, we remove the declaration, ‘~~The corresponding maximum field-effect mobilities (μ_{FE}) are $49.8 \text{ cm}^2 \cdot \text{V}^{-1} \cdot \text{s}^{-1}$, $74.8 \text{ cm}^2 \cdot \text{V}^{-1} \cdot \text{s}^{-1}$, and $65.5 \text{ cm}^2 \cdot \text{V}^{-1} \cdot \text{s}^{-1}$ for devices with 5 nm HfO₂, 10 nm HfO₂ and 30 nm Al₂O₃ gate stacks, respectively. When comparing the results for HfO₂ dielectrics, it suggests a decreased μ_{FE} with reduced EOT.~~’ in the revised manuscript.

Fig. R4. The electrical properties of MoS₂ FETs based on 5 nm HfO₂, 10 nm HfO₂, and 30 nm Al₂O₃ as dielectric layer. (a-c) Transfer curves of MoS₂ FET with $V_{\text{ds}}=1\text{V}$. (d-f) Corresponding device mobility distribution.

7) The authors claim low contact resistance of $0.59 \text{ k}\Omega \cdot \mu\text{m}$. Based on an extraction with channel lengths of 5-75 μm (Fig 2d). Usually, without going to small channel lengths in the sub-micron regime, the errorbars on the contact resistance extraction are very high. What is the fitting error that the authors obtain? And thus, how confident are they about this value?

Author response:

Thanks for the reviewer’s comments. As shown in Fig. R5b, we also fitted the data points and extracted the R_c based on the TLM method of the short channel device with L_{ch} from 50 nm to 2 μm , and the extracted R_c is around $0.62 \text{ k}\Omega \cdot \mu\text{m}$, which is very close to the R_c extracted from the long channel devices of $0.59 \text{ k}\Omega \cdot \mu\text{m}$ in Fig. R5a (also Fig. 2d in main text). For clarity, we also add the data of the R_c extraction based on the short channel devices as Fig. S6 in Supporting Information as highlighted in red.

Also, the fitting parameters are shown in inset of Fig. R5b with the intercept value ($2 \cdot R_c$) of $1.246 \pm 0.056 \text{ k}\Omega \cdot \mu\text{m}$. Thus, the fitting error is quite small. Also, based on the data point that the current density is $0.72 \text{ mA}/\mu\text{m}$ at $V_{ds}=1\text{V}$ of the short channel device with $L_{ch}=50 \text{ nm}$, thus, the total resistance (R_{tot}) is $1.39 \text{ k}\Omega \cdot \mu\text{m}$. So, the contact resistance R_c should be less than $R_{tot}/2=0.695 \text{ k}\Omega \cdot \mu\text{m}$.

We have added the R_c extraction based on short channel devices in Supporting information as Fig. S5 for clarity.

Fig.R5. The extraction of contact resistance is based on TLM method. (a) The R_c extraction based on long channel devices. (b) The R_c extraction based on short channel devices.

8) The argument on having a better device architecture due to lower water permeability in L211-219 while using uncapped devices is meaningless. I advise to remove this discussion if there is no further evidence of improved resistance against water permeability in capped devices.

Author response:

We totally agree with the reviewer’s comments. This argument was added per the previous reviewer’s request. We have removed this discussion about the water permeability in our manuscript.

9) Bending and strain: Authors state that the maximum strain is 2.5% which is very high, and it is doubtful that the dielectric and metals are not cracking at those strain values. Did the authors verify that strain is imparted correctly onto the device? Please provide Raman measurements of MoS_2 under strain to confirm that 2.5% is really encountered by the devices. Please also report on the gate leakage current upon straining because a cracked dielectric could lead to increased gate leakage.

Author response:

Thanks for the reviewer's comments. In our experiment, we applied strain on our flexible devices and measure the electrical properties. The setup for strain experiment is shown in Fig. R6. From the zoom in optical images of the devices under strain of ~2%, we could see that the device is completed without breaks or cracks.

Per the reviewer's suggestions, we also carry out further Raman measurements to check whether the strain is applied correctly as shown in Fig. R7. Based on the results, we could observe that the E_g mode has a red shift from $\sim 382 \text{ cm}^{-1}$ to 374 cm^{-1} with the bending radius changing from 12 mm to 3 mm, while the A_g mode positions roughly unchanged around 406 cm^{-1} . Based on the reference work (Li, Z., Lv, Y., Ren, L. *et al.* Efficient strain modulation of 2D materials via polymer encapsulation. *Nat Commun* **11**, 1151 (2020), <https://doi.org/10.1038/s41467-020-15023-3>), we could calculate the strain we applied on monolayer MoS_2 roughly changed from 0.2% to 1.6%, which is a little smaller than the expected values from 0.5% to 2% but follows well with the trend of the calculated strain based on formula $\epsilon = d/R$.

We also provide the leakage current in the Supporting Information as Fig. S8 when applying the strain on our device. We could see that the leakage current level keeps $<1 \text{ nA}$ when $R > 4 \text{ mm}$, and gradually increases to 3 nA with the strain when R approaches 2.4 mm as shown in Fig. R8 a. Also, we also provide the leakage current after bending 20 times, we could see that the leakage current increases to $\sim 20 \text{ nA}$ level. As shown in Fig. R8c, we could observe some tiny wrinkles/cracks formed on the dielectric layer after 20 times bending, which may induce the increased leakage current under strain.

Fig. R6. The strain measurement set up. a. The optical image of the set up for strain measurements. **b.** Optical image of the device under strain of ~2%.

Fig. R7. The strain analysis based on Raman characterizations of monolayer MoS_2 on PET substrate. a. Raman spectrum of monolayer MoS_2 with bending radius from 12 mm to 3 mm. **b.** The

statistical E_g and A_g mode position as a function of bending radius. **c.** The comparison between calculated strain ($\varepsilon = d/R$, where $2d$ is the thickness of the PET substrate) and measured strain based on Raman results. Note that the pristine Raman mode position was celebrated by the monolayer MoS_2 without strain.

Fig. R8. The leakage current analysis under strain. a. The leakage current recorded during the bending test with the bending radius from 12 mm to 2.4 mm. **b.** The leakage current increases to ~ 20 nA after bending 20 times. **c.** The optical image of the device after bending 20 times, where the red arrows indicate the tiny wrinkles.

10) As stated in L234, how do authors suggest to "adopting CMOS units as building blocks" in a purely n-type unipolar technology?

Author response:

Sorry for the misleading information here. Here, in our work we only adopt N-type MoS_2 as building blocks for constructing logic gates. We also propose a possibility like 'adopting CMOS units' for future flexible integrated electronics, such as combing P-type FETs like Cu-SnS_2 FETs, or bipolar type WSe_2 FETs, etc. For clarity, we revised the purpose as 'combing p-type FETs as building blocks' in our manuscript.

11) Fig 3c: The switching transition is below $V_{dd}/2$. As the authors write in L226-228: "The trip point of the inverter is when V_{out} equals V_{in} and of particular importance for low-power electronics as it determines the driving voltage (V_{dd}) of integrated logic circuits." The noise margin should be extracted in a measurement where V_{in} is swept up to the value of V_{dd} . Then it becomes obvious that the noise margins for high and low will be asymmetric. It is misleading to extract this with a V_{in} range not up to V_{dd} . As the authors show in Fig 3e, they operate the logic gates at $V_{in} = V_{dd} = 0.5$ V.

Author response:

We agree with the reviewer's comments here. The noise margin is much more important for logic functions. In our work, we just show the results with the highest gain value previously, however, the low driving voltage and noise margin is more important for realizing low-power electronics. Thus, we moved such data in supporting information Fig. S6, and added the data with ideal noise margin of with corresponding inverter gain of 120 in manuscript as shown in Fig. 3b and revised the manuscript correspondingly.

Fig. 3b. The noise margin and inverter gain of MoS₂ inverter.

12) How is this technology low power or energy efficient if a unipolar technology is used and a significant current ($I_{dd} > 0.4$ nA) flows at high V_{in} (Extended Data Fig 8f). Additionally, regarding the low power arguments: A threshold voltage averaging around 0V is not desirable, because the device should be fully off at $V_{GS} = 0$ V, requiring a positive V_T , good SS and low off current.

Author response:

We agree with the reviewer’s low power arguments here. In our work, we use the scaling down effect to realize the low power applications by adopting ultra-thin dielectric layer for logic circuit integrations based on single transistor units. The sub-1V driving voltage is quite desirable for low power electronics. For 0.4 nA driving current of MoS₂ inverter, the MoS₂ FETs have been working in the deep-subthreshold region. Thus, our technique has the potential for realizing low power electronics.

In order to compare the device uniformity, we include statistics on the pinch-off voltage ($V_{pinch-off}$) distribution at $I_{ds} = 10^{-7}$ mA level as shown in Fig. R9a. However, according to the definition of threshold voltage (V_{th}), the V_{th} has a positive value at 0.8V when the $V_{pinch-off}$ voltage is -0.3V as shown in Fig. R9b. Thus, in our manuscript, the MoS₂ FETs with close-to-zero $V_{pinch-off}$ voltage corresponds that they all feature with a positive V_{th} . Thus, we use the statistical method according to the definition of V_{th} and recalculated the distribution in the revised main text as ‘ V_{th} is centered at 0.96 ± 0.4 V’ in para 2, page 6.

Thus, based on our device data, the MoS₂ FETs feature with positive V_{th} , good SS and low off current for realizing low-power electronics. Note that we have revised the V_{th} statistic results based on the definition.

Further, as shown in Table S6, in order to compare the power consumption based on the unipolar transistors, we listed two representative ultra-low power works [ref. 34: Jiang, C. et al. Printed subthreshold organic transistors operating at high gain and ultralow power. *Science* **363**, 719-723 (2019); and ref 60: Lee, S. & Nathan, A. Subthreshold Schottky-barrier thin-film transistors with ultralow power and high intrinsic gain. *Science* **354**, 302-304 (2016)] for comparisons. We could see that our work features a power consumption of 408 pW for the typical device with channel length of $L = 10 \mu\text{m}$, and width of $W = 40 \mu\text{m}$, which is on the same order (130-600 pW) as literature works about the ultra-low power declaration. Thus, our ML-MoS₂ transistors, working in the deep-subthreshold region, have the feature of low-power consumption, which is essential for wearable or implantable devices application requirements.

Fig. R9. The pinch off voltage ($V_{\text{pinch-off}}$) and threshold voltage (V_{th}) comparison.

Table S6. Power consumption comparisons with representative ultra-low power transistor works.

Device structure	Inverter power comparisons
Science 354, 302 (2016)	
Science 363, 719 (2019)	
This work	

13) Extended Data Fig 3b: At which frequency was the MOSFET capacitance measured and what was the active area assumed? Due to channel accumulation the whole MoS₂ will be an active part of the parallel plate capacitor in accumulation and due to a parasitic MoS₂ resistance there could be an underestimated capacitance if measured at high frequencies. Did the authors see a pronounced frequency dispersion in the MOSFET-capacitors?

Author response:

Thank you for the reviewer's comments here. Fig. R10 a-b shows the capacitance characterizations of MoS₂ FET as a function of bias voltage from -2V to 4V with $f=10$ KHZ and as a function of operating frequency from 1 KHZ to 5000 KHZ with bias voltage of $V_{\text{bias}}=3$ V. The activate area is outlined by the black dash line on the optical image (inset in Fig. R10 a), which includes the metallic metal contact areas and semiconducting MoS₂ areas. We could see that the measured capacitance is nearly a constant when $f < 200$ KHZ and start to decrease at higher frequencies when $f > 200$ KHZ. Thus, that's correct that the capacitance would be underestimated only when the results measured at high frequencies ($f > 200$ KHZ).

Fig. R 10. The capacitance measurements of MoS₂ MOSFET capacitance. (a) The capacitance as a function of bias voltage with $f=10$ KHZ. (b) The capacitance as a function of operating frequency from 1 KHZ to 5000 KHZ with $V_{\text{bias}}=3$ V.

14) Ring oscillator frequency: The authors show good results in the ns-regime. To evaluate this result, they should clearly state the device dimensions and gate-to-drain/source overlaps. Their statement in line 256-257 "Note that the channel length/width of each TFT unit varies from 1 μm to 40 μm for the ROs." is unclear in this regard. In addition, I cannot find Supplementary Figure S3. Did the authors mean Figure S2 instead? Figure S2 is not sufficiently described to understand what kind of parasitic optimization was done. Please provide more details.

Author response:

Thanks for pointing these out. We have added the device details in our revised manuscript. Yes, it's Fig. S2 instead (previous version), sorry for the mistake here.

In our experiment, we optimized the device structure of each stage of MoS₂ ring oscillator to reduce the impact of parasitic capacitance on the frequency of a RO as shown in Supporting Information Fig. S7 a-b.

For the device in Fig. S7a, the channel length and channel width are $L=5 \mu\text{m}$, $w=40 \mu\text{m}$, individually. And the parasitic capacitance is roughly:

$$C_p(a)=2*(C_{\text{contact}}+C_{\text{MOS2_region}})=2*[10\ \mu\text{m}*(80+70)\ \mu\text{m}*2*C_{i_metal}+(100-45)\ \mu\text{m}*40\ \mu\text{m}*C_{i_MoS2}]=131.8\ \text{pF}$$

For the device in Fig. S7b, the channel length and channel width are $L=1\ \mu\text{m}$, $w=3\ \mu\text{m}$, individually. And the parasitic capacitance is greatly reduced:

$$C_p(b)=2*C_{\text{contact}}=2*(1\ \mu\text{m}*4\ \mu\text{m}*2*C_{i_metal})=0.205\ \text{pF}$$

The large parasitic capacitances in device (a) mainly come from the contact region/additional outer MoS₂ region; while the parasitic capacitance/per stage in device (b) is small due to the reduced overlapped region and the eliminated additional outer MoS₂ region. As a result, the corresponding experimentally achieved output frequencies are 1.491 MHz (a) and 49.4 MHz (b).

We have revised the manuscript as ‘*The typical channel length/width of TFT unit is 1 μm /3 μm for Ros, and more device details and parasitic capacitance optimization is demonstrated in Supporting Information Fig. S7*’ as highlighted in red.

Fig. S7. The oscillating frequency optimization of RO by reducing the parasitic capacitances from the contact electrodes/additional outer MoS₂ region. Up: the device design optimization of ring oscillators. Down: the output oscillating signals of ROs after optimizing the device design.

15) Ring oscillator benchmarking Fig 4f. The authors have missed crucial work on CNTs from literature which actually outperforms their circuits (<https://doi.org/10.1038/s41467-022-34621-x>). Please include this in the benchmark. Nevertheless, the here presented result seems to be a good improvement over previous flexible MoS₂ literature mostly in regard to operating voltage, as the authors have already achieved similar stage delays in their own prior work (<https://doi.org/10.1038/s41928-020-00475-8>).

Author response:

Thanks for the suggestions. We have added such benchmarks from CNTs to Fig. 4f as shown below. The CNT-data points are highlighted by blue triangle, while our data points are highlight in red. We could see that the CNT ROs have much smaller stage delay time, while our ROs could work under sub-1V supply voltage, which could demonstrate that low operating voltage is the main advantage of MoS₂-ROs. Also, we appreciate the reviewer’s evaluation of the current work.

Fig. 4f. Comparisons of propagation stage delay time as well as supply voltages of our work with literature results.

REVIEWER COMMENTS

Reviewer #5 (Remarks to the Author):

The reviewer thanks the authors for their great efforts in improving the manuscript providing additional data and clarifications. However, there are still some inconsistencies that need to be addressed before the work can be accepted for publications:

1) Correlation between grain size and mobility: The authors provide now benchmarking with grain size and mobility of MoS₂. The table indicates that grain size may be a factor, but it is doubtful that it is the major factor. Some single-domain mobility is lower than some good results with small grains. Could the authors also touch on the role of point defects and provide a more balanced discussion on contributing factors for high mobility in MoS₂?

2) The benchmarking was improved with regards to which results correspond to flexible electronics and which to rigid electronics. However, it doesn't seem to be fully consistent yet. Table S3 mentions an on-current of 936 $\mu\text{A}/\mu\text{m}$ but this was not demonstrated on flexible devices. The reviewer urges the authors to take care of consistent benchmarking with due diligence.

3) Fig 2e: The authors now marked there which results from them have been on rigid and which on flexible substrates. How is the situation about the grey data points from literature? Please make sure the readers can clearly understand where this data comes from and which is on flexible or rigid substrates.

4) Please add the additional data and discussions from the response letter to the supplement if not included yet. The reviewer thanks the authors for their additional work. Future readers would also benefit from these results. Thus please add discussion and Figs R2, R4, R6, R7 to the supplement as well.

5) Extended data Fig 5 still says that a radius of 2.4 mm corresponds to 2.5% strain, but the new data in Fig R7c suggests that only 1.6% could be verified by Raman spectroscopy. There might be slippage or another effect causing strain relaxation in the device compared to theoretically calculated values. In fact the measurement at 3 mm bending radius indicates that the strain does not increase with the same slope as predicted theoretically. Please revise accordingly.

6) Fig R8b: The caption says 20 nA, but it looks like that this might be the compliance setting the measurement setup (horizontal line between 4-5 V). This means leakage is in fact higher. Please revise the description accordingly.

7) C-V measurements: Thank you for clarifying, please also specify the frequency at which the data was acquired in Fig. S1 caption. Now that the capacitance is confirmed, the reviewer tried to verify the extracted mobility. Estimating on-current for the here claimed field-effect mobility of 70 cm^2/Vs (neglecting R_c), the reviewer calculates about twice the on-currents compared to what is presented in Fig 2e. Could you please clarify on your exact mobility extraction and confirm that it matches the on-currents measured? Otherwise, there could be a serious overestimation of mobility by a factor of 2.

8) Optimizing parasitics in the RO: Thank you for clarifying, please mention the stage number and stage delay in Fig S7 caption as well to improve clarity for the reader. Please label the additional MoS₂ region and overlaps in Fig S7 a and b if possible.

Point-by-point response letter

The reviewer thanks the authors for their great efforts in improving the manuscript providing additional data and clarifications. However, there are still some inconsistencies that need to be addressed before the work can be accepted for publications:

We sincerely appreciate the referee's great efforts on reviewing our manuscript and constructive comments here. The remaining inconsistencies and comments are addressed in our revised manuscript. Please see the detailed responses below.

1) Correlation between grain size and mobility: The authors provide now benchmarking with grain size and mobility of MoS₂. The table indicates that grain size may be a factor, but it is doubtful that it is the major factor. Some single-domain mobility is lower than some good results with small grains. Could the authors also touch on the role of point defects and provide a more balanced discussion on contributing factors for high mobility in MoS₂?

Author response:

We totally agree that, in addition to grain boundaries, point defects are another factor. From our experiences, the contribution to the quality degradation in MoS₂ from point defects are significant.

We have studied the density of point defects in our CVD MoS₂ samples. As shown in typical TEM result shown in Extended Data Fig. 1, we can see the defect density is rather low. Note that the TEM characterizations are local. In order to investigate the defect densities at the macro scale, we have developed a wet etching method to visualize those structural defects. Please refer to our newly published paper entitled 'Direct visualization of structural defects in 2D semiconductor' on *Chin. Phys. B* (2022). In this work, we have compared three types of samples, i.e., exfoliated MoS₂ flakes, small-grained (~few μm) CVD MoS₂ films, and large-grained (~few hundreds of μm) CVD MoS₂ films. Our statistical results reveal that the large-grained samples have much lower density of etched pits than the exfoliated samples (by 1 order of magnitude) and small-grained samples (by 2-3 order of magnitudes). The defect density characterizations tell us that the large-grained MoS₂ samples have the lowest density of both point defects and grain boundaries, providing the highest electronic quality.

We have added these related discussions in our revised Supplementary Information Section 1 as highlighted in red.

2) The benchmarking was improved with regards to which results correspond to flexible electronics and which to rigid electronics. However, it doesn't seem to be fully consistent yet. Table S3 mentions an on current of 936uA/um but this was not demonstrated on flexible devices. The reviewer urges the authors to take care of consistent benchmarking with due diligence.

Author response:

Thanks for pointing out the inconsistencies here. We agree with the referee's comments, and we have revised Table S3 as highlighted in red.

The maximum current density of our flexible FETs is $53.8 \mu\text{A} \cdot \mu\text{m}^{-1}$ at $V_g = 7 \text{ V}$ and $V_{ds} = 1 \text{ V}$ for the device with $L_{ch} = 5 \mu\text{m}$, the average on/off ratio is 5×10^7 and SS averages at $83 \text{ mV} \cdot \text{dec}^{-1}$. In the current version, the benchmarking parameters are all extracted from flexible devices. Please see Table S3 in our Supplementary Information file for details.

Table S3. Comparisons of flexible ML-MoS₂ devices.

Parameters	NE 2021, 4, 495	NE 2020, 3, 711 (Our Previous work)	This work
Maximum current densities	$466 \mu\text{A} \cdot \mu\text{m}^{-1}$ @ $V_g = 16 \text{ V}, V_{ds} = 1 \text{ V},$ $L_{ch} = 82 \text{ nm}$	$35 \mu\text{A} \cdot \mu\text{m}^{-1}$ @ $V_g = 80 \text{ V}, V_{ds} = 5 \text{ V},$ $L_{ch} = 6 \mu\text{m}$	$53.8 \mu\text{A} \cdot \mu\text{m}^{-1}$ @ $V_g = 7 \text{ V}, V_{ds} = 1 \text{ V},$ $L_{ch} = 5 \mu\text{m}$
ON/OFF ratio	2×10^6	$10^9 - 10^{10}$	5×10^7
Average mobility	$27 \sim 30 \text{ cm}^2 \cdot \text{V}^{-1} \cdot \text{s}^{-1}$	$\sim 55 \text{ cm}^2 \cdot \text{V}^{-1} \cdot \text{s}^{-1}$	$\sim 70 \text{ cm}^2 \cdot \text{V}^{-1} \cdot \text{s}^{-1}$
Subthreshold swings (SS)	$730 \sim 1000 \text{ mV} \cdot \text{dec}^{-1}$	$\sim 1000 \text{ mV} \cdot \text{dec}^{-1}$	$83 \text{ mV} \cdot \text{dec}^{-1}$
Pinch-off voltage distribution	$-12 \sim -8 \text{ V}$	$0 \sim 5 \text{ V}$	$-0.4 \sim +0.4 \text{ V}$

3) Fig 2e: The authors now marked there which results from them have been on rigid and which on flexible substrates. How is the situation about the grey data points from literature? Please make sure the readers can clearly understand where this data comes from, and which is on flexible or rigid substrates.

Author response:

Thanks for the referee's comments here. In our revised figure, we have separated the reference data points from rigid (in grey color) and flexible (in blue color) works as shown in Fig. 2e. Now Fig. 2e is much clearer in comparison with literature works.

Fig. 2e. Statistics of the on-state current density at $V_{ds}=1V$ versus channel length and comparisons with literature works. Please refer to Supplementary Table S2-S3 for parameter details.

4) Please add the additional data and discussions from the response letter to the supplement if not included yet. The reviewer thanks the authors for their additional work. Future readers would also benefit from these results. Thus, please add discussion and Figs R2, R4, R6, R7 to the supplement as well.

Author response:

We sincerely appreciate the referee's suggestion here. We have added the Figures (Figs R2, R4, R6, R7) into our revised Supplementary Information file as Fig. S1, S6, S10, S11 and related discussions as highlighted in red. Now the Supplementary Information is well-arranged for readers.

5) Extended data Fig 5 still says that a radius of 2.4 mm corresponds to 2.5% strain, but the new data in Fig R7c suggests that only 1.6% could be verified by Raman spectroscopy. There might be slippage or another effect causing strain relaxation in the device compared to theoretically calculated values. In fact, the measurement at 3 mm bending radius indicates that the strain does not increase with the same slope as predicted theoretically. Please revise accordingly.

Author response:

Thanks for the reviewer's comments here. In extended Data Fig. 5, we have revised the figure caption as 'd. Note that the bending radius of $R=3$ mm corresponds to the strain of $\epsilon=1.6\pm0.2$ % as verified by Raman spectrum in Supporting Information Fig. S11.' And we agree that the strain does not increase with the same slope as predicted when bending radius $R<3$ mm.

6) Fig R8b: The caption says 20 nA, but it looks like that this might be the compliance setting the

measurement setup (horizontal line between 4-5 V). This means leakage is in fact higher. Please revise the description accordingly.

Author response:

Thanks for the comments here. We have revised the description in Supplementary Information Fig. S12b as ‘The leakage current increases over than the compliance setting of ~20 nA after bending 20 times’ as highlighted in red.

Fig. S12. b. The leakage current increases over than the compliance setting of ~20 nA after bending 20 times.

7) C-V measurements: Thank you for clarifying, please also specify the frequency at which the data was acquired in Fig. S1 caption. Now that the capacitance is confirmed, the reviewer tried to verify the extracted mobility. Estimating on-current for the here claimed field-effect mobility of 70 cm²/Vs (neglecting R_c), the reviewer calculates about twice the on-currents compared to what is presented in Fig 2e. Could you please clarify your exact mobility extraction and confirm that it matches the on-currents measured? Otherwise, there could be a serious overestimation of mobility by a factor of 2.

Author response:

Thanks for the reviewer’s comments here. For the data acquired in Fig. S1b (now Fig S2b in the revised Supplementary Information), the test frequency is 100 KHZ based on the MOS and MIM structure. We have added the frequency information in figure caption as highlighted in red.

Fig. R1a is a typical transfer curve of a MoS₂ FET at V_{ds}= 1V with L_{ch}=5 μm and W=30 μm, and thus the corresponding current densities is 55.4 μA·μm⁻¹. Fig. R1b shows the calculated transconductance and the maximum value is 0.8786 mA/V at V_g=7V.

Based on the measured MOSFET capacitance result in Fig. S2b, the normalized capacitance is ~2.7×10⁻² F·m⁻² for 5 nm HfO₂ at V_g=4V. Thus, we could roughly calculate the normalized capacitance of 10 nm HfO₂:

$$C_i(10\text{ nm}) = \epsilon_0 \cdot \epsilon_r / d = 0.5 \times C_i(5\text{ nm}) = 1.35 \times 10^{-2} \text{ F} \cdot \text{m}^{-2}$$

and thus, the dielectric constant is $\epsilon_r(10\text{ nm}) = 15.2$ based on MOSFET structure. This value is roughly consistent with the measured dielectric constant of 14.5 of 10 nm HfO_2 deposited at 150 °C based on the MIM structure, as shown in Fig. 1b.

Thus, the field effect mobility for this typical device is:

$$\begin{aligned} \mu_{FE}(V_g = 7V) &= \frac{dI_{ds}}{dV_g} \cdot \frac{L}{W} \cdot \frac{1}{C_i \cdot V_{ds}} \\ &= 0.8786 \frac{\text{mA}}{\text{V}} \times \frac{5\mu\text{m}}{30\mu\text{m}} \times \frac{1}{1.35 \times 10^{-2} \text{F} \cdot \text{m}^{-2}} = 108.5 \text{cm}^{-2} \text{V}^{-1} \text{s}^{-1} \end{aligned}$$

Based on the same calculation process, the statistical mobility distribution extracted from the data in Fig. 2b are summarized in Fig. 2c, where the mobility averages at $\sim 70 \text{cm}^{-2} \cdot \text{V}^{-1} \cdot \text{s}^{-1}$, and the value of $108.5 \text{cm}^{-2} \cdot \text{V}^{-1} \cdot \text{s}^{-1}$ locates at the upper boundary.

Fig. S2. The capacitance/dielectric constant measurements of HfO_2 films. b. Capacitance-voltage (C-V) measurements of 5 nm HfO_2 with a frequency of 100 KHZ based on MOS and MIM structure.

Fig. R1. The device mobility extraction. (a) Typical transfer curves of a MoS_2 FET at $V_{ds}=1\text{ V}$ with $L_{ch}=5\text{ }\mu\text{m}$, $W=30\text{ }\mu\text{m}$, and $t_{\text{HfO}_2}=10\text{ nm}$. (b) The corresponding transconductance.

8) Optimizing parasitics in the RO: Thank you for clarifying, please mention the stage number and stage delay in Fig S7 caption as well to improve clarity for the reader. Please label the additional MoS₂ region and overlaps in Fig S7 a and b if possible.

Author response:

Thanks for the referee's suggestions here. For the parasitic capacitance optimization section, we use 3-stage ring oscillator as examples for demonstration, and corresponding stage delay is 111.8 ns (a) and 3.4 ns (b), individually. We have added the related information to Fig. S9 caption. Also, we highlight the overlapped region from the additional MoS₂ and contact electrodes as outlined by blue shadow.

Fig. S9. The oscillating frequency of 3-stage RO by reducing the parasitic capacitances from the contact electrodes/additional outer MoS₂ region. Up: the device design optimization of ring oscillators. Down: the output oscillating signals of ROs after optimizing the device design. Corresponding state delay is 111.8 ns (a) and 3.4 ns (b), individually.

REVIEWERS' COMMENTS

Reviewer #5 (Remarks to the Author):

The authors have fully addressed all my remaining comments. The work is ready for publication.